# LEARNING TO REGISTER UNBALANCED POINT PAIRS

## ABSTRACT

Point cloud registration methods can effectively handle large-scale, partially overlapping point cloud pairs. Despite its practicality, matching the unbalanced pairs in terms of spatial extent and density has been overlooked and rarely studied. We present a novel method, dubbed UPPNet, for **U**nbalanced **P**oint cloud **P**air registration. We propose to incorporate a hierarchical framework that effectively finds inlier correspondences by gradually reducing search space. The proposed method first predicts subregions within target point cloud that are likely to be overlapped with query. Then following super-point matching and fine-grained refinement modules predict accurate inlier correspondences between the target and query. Additional geometric constraints are applied to refine the correspondences that satisfy spatial compatibility. The proposed network can be trained in an end-to-end manner, predicting the accurate rigid transformation with a single forward pass. To validate the efficacy of the proposed method, we create a carefully designed benchmark, named KITTI-UPP dataset, by augmenting the KITTI odometry dataset. Extensive experiments reveal that the proposed method not only outperforms state-of-the-art point cloud registration methods by large margins on KITTI-UPP benchmark, but also achieves competitive results on the standard pairwise registration benchmark including 3DMatch, 3DLoMatch, ScanNet, and KITTI, thus showing the applicability of our method on various datasets. The source code and dataset will be publicly released.

## 1 INTRODUCTION

Point cloud registration is a task that aims to recover 3D rigid transformation between two possibly overlapping point cloud fragments. The rapid advance of commodity 3D sensors gives rise to the necessity of efficient point cloud registration algorithms for numerous real-world applications, including 3D reconstruction for virtual-, augmented-, and mixed reality applications, and the navigation systems of autonomous vehicles or robotic agents. Recent work has made remarkable progress in developing learning-based point cloud registration algorithms for tackling real-world 3D scans (Geiger et al., 2012; Zeng et al., 2017) with high-resolution feature extraction (Choy et al., 2019b; Bai et al., 2020) under presence of low ratio of the inlier correspondences (Choy et al., 2020b; Bai et al., 2021; Lee et al., 2021) or small overlap region between point pairs (Huang et al., 2021).

However, the imbalance issue in terms of *spatial extent* and *point density* between the input point clouds is often overlooked, despite its practical utility in the problems such as incremental mapping, or the registration of partial observations and the holistic environment. For instance, there are sensible solutions for registering a pair of 3D LiDAR scans, but registering a single LiDAR scan and a large-scale 3D map still remains challenging. A viable solution is to apply a global localization approach (Uy & Lee, 2018; Komorowski, 2021; Du et al., 2020; Zhang & Xiao, 2019; Liu et al., 2019), but the existing methods cast the problem as a retrieval task and assume that the 3D map is given as a set of overlapping 3D scans rather than a holistic map, which is not generally applicable to the unbalanced point pairs.

Recent feature-based pairwise point cloud registration methods are equipped with matchability detection (Bai et al., 2020), overlap detection (Huang et al., 2021), or hierarchical correspondence prediction (Yu et al., 2021), which are possibly advantageous in registering unbalanced point clouds. However, we empirically found that they collapse in registering unbalanced point clouds. Point cloud description and matching in the modern feature-based registration methods tend to be distracted by similar geometric structures that often appear in the larger point cloud.

To this end, we propose **UPPNet**, the first neural architecture which is designed to be efficient for large-scale **U**nbalanced **P**oint cloud **P**air registration. UPPNet is a hierarchical framework that effectively finds inlier correspondences by gradually reducing search space. In the coarsest level, a *submap proposal* module proposes the subregions that are likely to be overlapped with the query by utilizing a global geometric context. Then, the coarse-to-fine matching module predicts accurate point-level correspondences by utilizing attention-based context aggregation and solving optimal transport problems. The subsequent structured matching module filters out outlier correspondences that violate spatial compatibility.

To evaluate our method, we create a carefully designed benchmark, namely KITTI-UPP dataset, for matching point cloud pairs under the diverse spatial extent and point density imbalance by augmenting the KITTI odometry dataset (Geiger et al., 2012). The experiment shows that our method improves the *Registration Recall* on the KITTI-UPP dataset by over 19.6% than state-of-the-art registration pipelines when the target point cloud is 11.1 times spatially larger and 11.7 times denser than the query point cloud. Furthermore, we evaluate the proposed method under the unbalanced indoor environments using ScanNet (Dai et al., 2017) dataset and show that the proposed method can be generalized for indoor RGB-D scans. Finally, to demonstrate the applicability of the proposed method for partially overlapped point cloud pairs, we evaluate our method on the standard pairwise registration benchmarks, 3DMatch, 3DLoMatch (Zeng et al., 2017; Huang et al., 2021), and KITTI odometry (Geiger et al., 2012) datasets. The proposed method achieves competitive registration accuracy with the modern pairwise registration methods (Bai et al., 2020; Huang et al., 2021; Yu et al., 2021). An overview of our method can be found in Figure 1.

Our main contributions are summarized as follows:

- We propose a novel hierarchical framework that gradually reduces the search space via submap proposal module and coarse-to-fine matching modules, which can effectively handle unbalanced point cloud registration tasks.
- We introduce a new benchmark, namely KITTI-UPP dataset, by carefully augmenting the large-scale outdoor LiDAR dataset (Geiger et al., 2012).
- Our method can be trained in an end-to-end manner and demonstrates strong generalization ability on a wide range of spatial and point density. It outperforms the previous state-of-the-art methods by 19.6% *Registration Recall* on a challenging benchmark.
- Our method achieves competitive registration accuracy in both unbalanced indoor RGB-D registration (Dai et al., 2017) and the standard pairwise registration benchmarks (Zeng et al., 2017; Huang et al., 2021; Geiger et al., 2012), showing the applicability of the method.

## 2 RELATED WORK

**Point cloud registration.** Given partially overlapping point cloud pairs, point cloud registration aims to estimate the rigid transformation parameters that align the input point clouds. Typical pipelines start with the feature extraction and matching stage to produce a set of putative correspondences, followed by a robust model fitting algorithm to estimate the pose parameters from given correspondences.

Traditional local descriptors for point clouds (Johnson & Hebert, 1999; Rusu et al., 2008; 2009; Tombari et al., 2010; Salti et al., 2014) encode local geometry using hand-crafted features such as surface normal or curvature. Recent learnable feature descriptors train a network to extract geometric features in a data-driven manner. Qi et al. (2017) incorporate shared MLP and permutation-invariant aggregation operations to process unordered and irregular point cloud data. Deng et al. (2018a;b) extended Qi et al. (2017) by combining point pair feature to extract global context-aware feature descriptors. Choy et al. (2019b) adopted the fully convolutional networks to process point cloud data by utilizing sparse convolution (Choy et al., 2019a) and achieved state-of-the-art feature matching accuracy in large-scale real-world datasets. Bai et al. (2020) and Huang et al. (2021) incorporate keypoint and overlap confidence prediction for robust feature matching between low overlapping point pairs.

The set of putative correspondences that is built by matching local feature descriptors tends to contain a high portion of outliers. Hence, the pose estimation algorithm should be robust against the existence of outliers. RANdom SAmple Consensus (RANSAC) (Fischler & Bolles, 1981) and

its variants (Fitzgibbon, 2003; Raguram et al., 2012; Hoseinnezhad & Bab-Hadiashar, 2011; Chum & Matas, 2005) are one of the most popular methods. Recent studies of learning-based outlier rejection methods apply for the problem of two-view correspondences (Moo Yi et al., 2018; Zhang et al., 2019; Brachmann et al., 2017) and 3D correspondences (Choy et al., 2020b;a). Lee et al. (2021) utilizes Hough voting in 6D parameter space and learnable refinement module, resulting in a robust and efficient registration pipeline for the large-scale point cloud. Bai et al. (2021) incorporate the spatial compatibility to filter out noisy correspondences. Yu et al. (2021) proposed to avoid keypoint detection by incorporating hierarchical correspondence extraction modules and Lu et al. (2021) present specialized pipeline for large-scale LiDAR scans. However, none of those methods are explicitly designed for the unbalanced point cloud registration and we empirically found that those methods collapse under the extreme imbalance in terms of spatial extent and point density.

**Global localization.** The early global localization algorithms are based on the 2D images (Chen et al., 2017; Sarlin et al., 2019; Sattler et al., 2017), where input images are matched against the 3D map reconstructed with Structure from Motion (SfM). They often cast a visual localization task as a retrieval problem, where the query images are described with global descriptors, and the most similar point features are retrieved from the database. Vector of Locally Aggregated Descriptors (VLAD) (Jégou et al., 2010; Arandjelovic & Zisserman, 2013) is one of the most widely used approaches. It aggregates the local descriptors into a single global descriptor via clustering and is designed to be very low dimensional. NetVLAD (Arandjelovic et al., 2016) has extended it to learnable method.

Recently, 3D point cloud-based global localization methods are developed. Yin et al. (2017; 2018) projects a point cloud onto a spherical image and then extracts a global feature descriptor via 2D CNN. In Yin et al. (2019), it is combined with a localization module, which utilizes particle filter and ICP for 3D pose estimation. Uy & Lee (2018) apply NetVLAD to point cloud data using PointNet Qi et al. (2017) as backbone network. Based on this, Zhang & Xiao (2019) employs per-point attention to enhance the global descriptor. Liu et al. (2019) applies a graph neural network on hand-crafted point features to encode local context and aggregate it into global descriptors using the NetVLAD layer. Du et al. (2020) jointly optimizes both local and global feature extractors and solves both global retrieval and local registration in a single forward pass. Komorowski (2021) utilizes sparse convolutional networks to overcome the limitation of the local context of PointNet.

One feasible solution to tackle the unbalanced point cloud registration is to device the task into two separate sub-tasks; 1) the nearest frame retrieval and 2) local registration. After the target frame is retrieved, the relative pose between the query and the target frame is predicted by applying the conventional pairwise registration methods. These methods, however, suffer from several drawbacks. First, there are significant overheads in both the computational cost and the memory footprint to represent a scene with a set of multiple overlapping local frames. Second, they are likely not to generalize when point cloud pairs have a domain difference. For instance, considering the registration between the reconstructed point cloud of a large-scale scene and a local scan, the global localization methods fail to be directly adopted due to the severe density difference. Finally, the success of the registration strongly depends on the localization results. They can not recover when the coarse alignment via localization fails.

## 3 METHOD

This section describes the proposed UPPNet, designed for registering unbalanced point cloud pairs. To handle spatial and density imbalance, we introduce a hierarchical framework consisting of three levels of matching stages, submap matching (Section 3.3), super point matching (Section 3.4), and point matching (Section 3.5). The overview of the proposed pipeline is illustrated in Figure 1.

### 3.1 PROBLEM DEFINITION

Given a pair of possibly overlapping point clouds, $\mathbf{X} \in \mathbb{R}^{n \times 3}, \mathbf{Y} \in \mathbb{R}^{m \times 3}$, our goal is to estimate the optimal rigid transformation $\mathbf{R}^*, \mathbf{t}^*$ that minimizes the geometric error defined as follows:

$$\mathbf{R}^*, \mathbf{t}^* = \arg\min_{\mathbf{R}, \mathbf{t}} \sum_{(i,j) \in \mathcal{C}} \|\mathbf{R}\mathbf{y}_i + \mathbf{t} - \mathbf{x}_j\|_2 \tag{1}$$

where $\mathbf{R} \in SO(3), \mathbf{t} \in \mathbb{R}^3$ are rotational matrix and translation vector, $\mathcal{C}$ is a set of true correspondences between $\mathbf{X}$ and $\mathbf{Y}$. In this paper, we are interested in the case where $\mathcal{V}(\mathbf{X}) \gg \mathcal{V}(\mathbf{Y})$, and

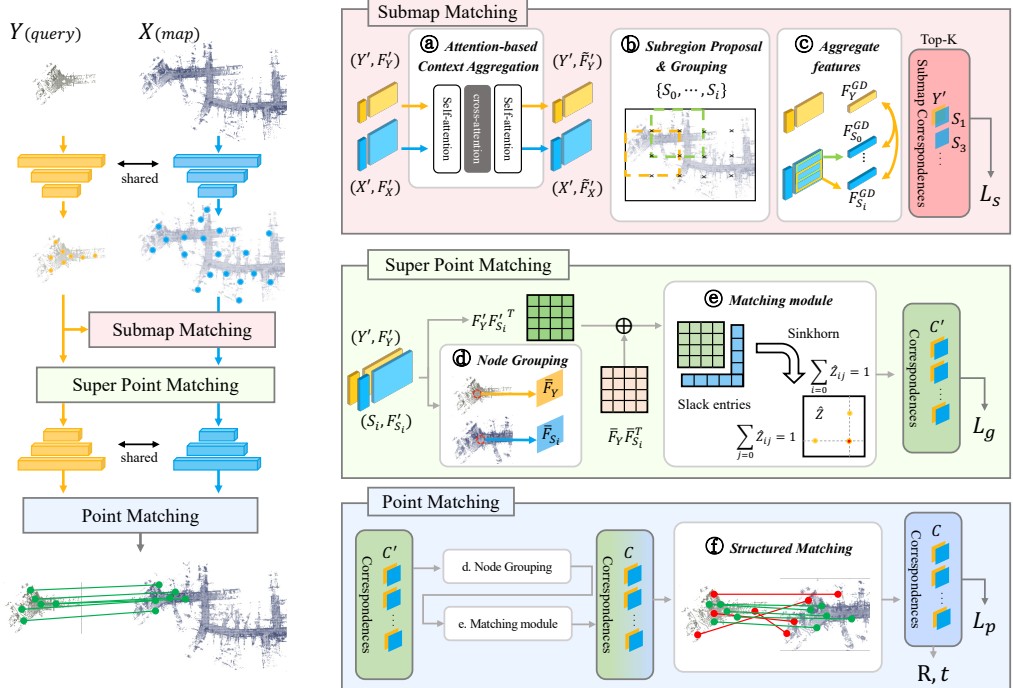

Figure 1: (Left) Overview of UPPNet. Given an unbalanced point cloud pair, UPPNet hierarchically reduce the search space by utilizing multi-level features. (Top right) Super point features get strengthen via *Attention-based Context aggregation module*, shown at ⓐ, and aggregated into global descriptor using Generalized mean pooling, shown at ⓒ. We select the submaps with top-k similarity. (Middle right) We build a similarity matrix using super point features in each selected submap. We solve the optimal transport problem on the similarity matrix, and the super point correspondences are estimated, shown at ⓔ. (Botton right) Super point correspondences are refined to point correspondences by utilizing node grouping, shown at ⓓ, and matching module. ⓕ Structured matching filters out the noisy correspondences that do not satisfy spatial compatibility. Modules in UPPNet, including feature extraction and correspondence estimation, can be trained in an end-to-end manner with the three losses $\mathcal{L}_s$, $\mathcal{L}_g$, and $\mathcal{L}_p$.

$n \gg m$, where $\mathcal{V}(\cdot)$ denotes the spatial volume of the bounding box that tightly covers the input point cloud. In other words, there is spatial and density imbalance between $\mathbf{X}$ and $\mathbf{Y}$. From here on, we will refer to the reference point cloud that spans a larger spatial extent with higher density ($\mathbf{X}$) as a *map*, and a query point cloud with a small and sparse point cloud ($\mathbf{Y}$) as a *query*. Furthermore, we use the terms **spatial imbalance factor**, $\rho_\mathrm{s} = \frac{\mathcal{V}(\mathbf{X})}{\mathcal{V}(\mathbf{Y})}$, and **point imbalance factor**, $\rho_\mathrm{p} = \frac{n}{m}$, to denote the relative imbalances between two point clouds in terms of spatial extent and number of points. Note that we can also calculate the **density imbalance factor** by $\rho_d = \frac{\rho_p}{\rho_s}$.

## 3.2 FEATURE EXTRACTION

The proposed UPPNet begins with the feature extraction to encode the geometric context of map and query point clouds. To achieve this goal, we adopt a shared U-shaped network implemented with KPConv (Thomas et al., 2019) to extract the multi-level features. Given an input point cloud $\mathbf{X} \in \mathbb{R}^{n \times 3}$ with initial feature $\mathbf{F}_\mathbf{X}^\mathrm{in} \in \mathbb{R}^{n \times d_\mathrm{in}}$, the feature extractor $f_\theta(\cdot)$ outputs the pointwise feature $\mathbf{F}_\mathbf{X} \in \mathbb{R}^{n \times d}$ that encode local geometric context, denoted as $(\mathbf{X}, \mathbf{F}_\mathbf{X}) = f_\theta(\mathbf{X}, \mathbf{F}_\mathbf{X}^\mathrm{in})$. At the most coarse resolution, i.e., at the end of the encoder, we obtain downsampled points with corresponding feature vectors, and we call the downsampled points as **super points** and denotes the point coordinates and the features with $\mathbf{X}' \in \mathbb{R}^{n' \times 3}$ and $\mathbf{F}_{\mathbf{X}'}$. For each super point $\mathbf{x}_i'$, the input point cloud $\mathbf{X}$ can be partitioned into groups by assigning each point to its closest super point as in Yu et al. (2021):

$$\mathbf{G}_i = \{\mathbf{x} \in \mathbf{X} \,\big|\, \|\mathbf{x} - \mathbf{x}_i'\| \leq \|\mathbf{x} - \mathbf{x}_j'\|, \forall j \neq i]\}, \tag{2}$$

where we denote the corresponding super point feature map as $\mathbf{F}'_{\mathbf{G}_i}$.

## 3.3 SUBMAP MATCHING

**Building submaps.** The first stage of our hierarchical registration pipeline is the *submap proposal* that builds the submap candidates that are likely to be overlapped with the query point cloud. To do so, we divide the map into $L$ overlapping submaps. A submap $\mathbf{S}_i$ is a subset of superpoints of the map, i.e., $\mathbf{S}_i \subset \mathbf{X}'$, that lie in $i$-th submap region $\mathcal{S}_i$: $\mathbf{S}_i = \{\mathbf{x}' \in \mathbf{X}' : \mathbf{x}' \in \mathcal{S}_i\}$. Each (cubic-shaped) submap region $\mathcal{S}_i$ with its center $\mathbf{c}_i \in \mathbb{R}^3$ (which we call center point) and edge length of $v \in \mathbb{R}^+$ is formally defined as $\mathcal{S}_i = \{\mathbf{s} \in \mathbb{R}^3 : \max|\mathbf{s} - \mathbf{c}_i| \leq v/2\}$. In this work, we evenly place $L$ overlapping submap regions $\{\mathcal{S}_i\}_{i=1}^L$ to cover all the superpoints of the map $\mathbf{X}'$ with carefully chosen overlapping ratio $\mu$, satisfying $|\mathcal{S}_i \cap \mathcal{S}_j|/|\mathcal{S}_i| = \mu$ and $||\mathbf{c}_i - \mathbf{c}_j|| = v \cdot \mu$ for any adjacent, overlapping submap regions $\mathcal{S}_i$ and $\mathcal{S}_j$. We set $\mu = 0.5$ in our experiments. The edge length of the submap regions, $v \in \mathbb{R}^+$, is defined by the spatial size of the query $\mathbf{Y}$: Specifically, we first compute the furthest distance between two points in $\mathbf{Y}$ along $i$-th axis as $s_i = |\max_a \mathbf{y}_{a,i} - \min_b \mathbf{y}_{b,i}| \in \mathbb{R}^+$, thus giving $\{s_i\}_{i=1}^3$ of which respective elements represent width ($s_1$), height ($s_2$), and length ($s_3$) of $\mathbf{Y}$. The edge length of submap regions is defined as the largest edge length of $\mathbf{Y}$: $v = \max_i s_i$. The overview of the submap proposal and grouping is shown in Figure 1 (b).

**Attention-based context aggregation.** The super point feature maps of query and submaps are then strengthened via *attention-based context aggregation* module (Yu et al., 2021; Huang et al., 2021; Sarlin et al., 2020) to incorporate the global geometric context. As in Huang et al. (2021), we construct a bipartite graph between query and map point cloud regarding the super points as nodes, and we apply a sequence of self-, cross-, and self-attention layers to fuse the global context between two point clouds and augment each super point feature. Given super point feature maps $(\mathbf{F}'_{\mathbf{X}}, \mathbf{F}'_{\mathbf{Y}})$, we linearly project the source feature $\mathbf{F}'_{\mathbf{X}}$ to *query*, $\mathbf{Q} = \mathbf{W}_{\mathbf{Q}} \mathbf{F}'_{\mathbf{X}}$ and target feature $\mathbf{F}'_{\mathbf{Y}}$ to *key* and *value* as $\mathbf{K} = \mathbf{W}_{\mathbf{K}} \mathbf{F}'_{\mathbf{Y}}, \mathbf{V} = \mathbf{W}_{\mathbf{V}} \mathbf{F}'_{\mathbf{Y}}$, respectively, where $\mathbf{W}_{\mathbf{Q}}, \mathbf{W}_{\mathbf{K}}$, and $\mathbf{W}_{\mathbf{V}}$ are the learnable parameters. To calculate the message $\mathbf{M}$ that flows in the graph, we use *attention-based aggregation* as follows: $\mathbf{M} = \frac{\mathbf{Q}\mathbf{K}^T}{\sqrt{b}} \cdot \mathbf{V}$, where $b$ is the channel dimension of super point features. When calculating self-attention messages, we take $(\mathbf{F}'_{\mathbf{X}}, \mathbf{F}'_{\mathbf{X}})$ and $(\mathbf{F}'_{\mathbf{Y}}, \mathbf{F}'_{\mathbf{Y}})$ as input and flow messages to corresponding subgraphs.

**Global feature aggregation.** After we augment super point features, we aggregate super point features into a global descriptor for each submap. The super point features are lifted into a higher dimension by a linear layer and then aggregated into a global descriptor via Generalized Mean Pooling (GeM) (Radenović et al., 2018). GeM has shown its strength in localization tasks for various modality (Tolias et al., 2015; Komorowski, 2021). Concisely, GeM aggregates the super point features $\mathbf{F}'_{\mathbf{S}_i}$ for submap $\mathbf{S}_i$ and builds the global descriptor $\mathbf{F}^{\mathrm{GD}}_{\mathbf{S}_i}$ as $\mathbf{F}^{\mathrm{GD}}_{\mathbf{S}_i}(k) = \left( \frac{1}{|\mathbf{S}_i|} \sum_{\mathbf{x}'_j \in \mathbf{S}_i} (\mathbf{F}'_{\mathbf{x}'_j}(k))^\alpha \right)^{1/\alpha}$, where $\mathbf{F}^{\mathrm{GD}}_{\mathbf{S}_i}(k)$ is $k$-th element of the global descriptor, and $\alpha$ is a learnable parameter.

**Global feature matching.** With the aggregated global descriptors for each submap $\{\mathbf{F}^{\mathrm{GD}}_{\mathbf{S}_i}\}$ and query $\mathbf{F}^{\mathrm{GD}}_{\mathbf{Y}}$, we calculate the similarity matrix using L2 distance between two global descriptors:

$$d^{\mathrm{GD}}_i = \|\mathbf{F}^{\mathrm{GD}}_{\mathbf{S}_i} - \mathbf{F}^{\mathrm{GD}}_{\mathbf{Y}}\|_2, \quad \mathbf{d}^{\mathrm{GD}} = \begin{bmatrix} d^{\mathrm{GD}}_1 \cdots & d^{\mathrm{GD}}_m \end{bmatrix} \in \mathbb{R}^{m \times 1}, \tag{3}$$

where $\mathbf{d}^{\mathrm{GD}}$ is a one dimensional vector because the query point cloud $\mathbf{Y}$ is described with a single global descriptor $\mathbf{F}^{\mathrm{GD}}_{\mathbf{Y}}$, whereas the map $\mathbf{X}$ is described with $m$ global descriptors $\mathbf{F}^{\mathrm{GD}}_{\mathbf{S}_i}$ for each submap $\mathbf{S}_i$. On train time, we apply a loss function on $\mathbf{d}^{\mathrm{GD}}$ with groundtruth supervision. On inference time, we pick $k$ submaps with the top-$k$ similarity values, where $k$ is a hyperparameter.

## 3.4 SUPER POINT MATCHING

After the candidate submaps are proposed with the consideration of the global geometric context, we perform the super point matching. For each proposed submap, the super points belonging to the submap $\mathbf{S}_i$ are retrieved with corresponding super point features, $\mathbf{F}'_{\mathbf{S}_i} \in \mathbb{R}^{|\mathbf{S}_i| \times d'}$. Then we leverage super point features $\mathbf{F}'_{\mathbf{S}_i}, \mathbf{F}'_{\mathbf{Y}}$ of submap and query to calculate the similarity matrix $\mathcal{S}_{\mathbf{S}_i}$ using inner product:

$$\mathcal{S}_{\mathbf{S}_i} = \mathbf{F}'_{\mathbf{S}_i} \mathbf{F}'^T_{\mathbf{Y}}, \quad \mathcal{S}_{\mathbf{S}_i} \in \mathbb{R}^{|\mathbf{S}_i| \times n'} \tag{4}$$

**Neighborhood constraint.** The similarity matrix $\mathcal{S}_{\mathbf{S}_i}$ effectively convey the geometric information with moderate receptive field size. Following Lu et al. (2021), we incorporate additional geometric context, called neighborhood constraint, to handle challenging cases. As described in Eq. 2, each point $\mathbf{x} \in \mathbf{X}$ are assigned to the closest super point. For each super point $\mathbf{x}'_i \in \mathbf{X}'$, we aggregate the neighbor features of points within the partition $\mathbf{G}_i$ using max pooling and denote the aggregated features as *neighbor feature* ,$\bar{\mathbf{F}}_{\mathbf{G}_i}$. Now we calculate another similarity matrix $\mathcal{N}_{\mathbf{S}_i} = \bar{\mathbf{F}}_{\mathbf{S}_i}^T \bar{\mathbf{F}}_{\mathbf{Y}}$, called *neighbor similarity matrix*. The final similarity matrix is defined as addition of two similarity matrices augmented with additional row and column for slack entries to handle unmatched super points as suggested in Yu et al. (2021); Sarlin et al. (2020):

$$\mathcal{Z}_{\mathbf{S}_i} = \begin{bmatrix} \mathcal{S}_{\mathbf{S}_i} + \mathcal{N}_{\mathbf{S}_i} & \mathbf{z} \\ \mathbf{z} & z \end{bmatrix}, \quad \mathcal{Z}_{\mathbf{S}_i} \in \mathbb{R}^{(|\mathbf{S}_i|+1) \times (n'+1)} \tag{5}$$

We then utilize Sinkhorn algorithm (Sinkhorn & Knopp, 1967) to solve optimal transport problem on $\mathcal{Z}_{\mathbf{S}_i}$ and obtain $\bar{\mathcal{Z}}_{\mathbf{S}_i}$. Each entry $(i, j)$ in $\bar{\mathcal{Z}}_{\mathbf{S}_i}$ indicates normalized probability whether $(i, j)$ correspondence is a true correspondence. By thresholding $\bar{\mathcal{Z}}_{\mathbf{S}_i}$ by predefined value $\tau_{\mathcal{Z}}$, we obtain the set of predicted super point correspondences, $\mathcal{C}'$, and they passed to the final point matching module to point correspondences, $\mathcal{C}$.

**Structured matching.** We use spatial compatibility (Bai et al., 2021; Lee et al., 2021) to further reject the outlier correspondences. It considers two correspondences $(p_i, q_i)$ and $(p_j, q_j)$ are likely to be inliers if their spatial distance $|d(p_i, p_j) - d(q_i, q_j)|$ is smaller than predefined threshold. Extending this concept, for each correspondence, we compute the compatibility score with respect to the number of their compatible correspondences in the whole correspondence set. For example, whole correspondences set is $\{c1, c2, c3\}$ and compatibility score of $c1$ will be 2 when $c2$ and $c3$ satisfy spatial compatibility with $c1$. With the compatibility score, we consider correspondences with a low score as outliers and filter out them. In our hierarchical framework, we apply this strategy to our super point matching module and point matching module to extract only reliable correspondence pairs. Furthermore, KNN search is used to guarantee as many inliers as possible for the initial correspondence set. For more details about the structured matching, please refer to Appendix.

### 3.5 POINT MATCHING

Given predicted super point correspondence, we refine it to point-level correspondences for the final rigid transformation parameter estimation. We expand a single super point correspondence to a pair of point patches with neighboring points around the super points in each point cloud. We use the point-to-super point grouping as described in Eq. 2 and pass the selected point groups to the matching module as in Section 3.4. After applying the threshold to the similarity matrix and structured matching, we get the final set of point-level correspondences between two input point clouds. We then run RANSAC on the correspondence set to estimate the rigid transformation parameter. Note that there are multiple sets of correspondences since we perform the matching module for each proposed submap in parallel. We select the one with the highest inlier ratio among the multiple transformation candidates as our final prediction.

### 3.6 LOSS

We train our network using the loss function defined as $\mathcal{L} = \mathcal{L}_s + \lambda_g \mathcal{L}_g + \lambda_p \mathcal{L}_p$, where the total loss is the weighted sum of submap matching loss $\mathcal{L}_s$, super point matching loss $\mathcal{L}_g$, and point matching loss $\mathcal{L}_p$. Specifically, we define point matching loss $\mathcal{L}_p$ as:

$$\mathcal{L}_p = \frac{-\sum_{i,j} \hat{\mathcal{Z}}(i,j) \log \bar{\mathcal{Z}}(i,j)}{\sum_{i,j} \hat{\mathcal{Z}}(i,j)} \tag{6}$$

where $\bar{\mathcal{Z}}$ is the predicted similarity matrix after solving optimal transport using Sinkhorn algorithm, $\hat{\mathcal{Z}}$ is the binary matrix that indicates groundtruth. If $(i, j)$ correspondence is a true correspondence then $\hat{\mathcal{Z}}(i, j) = 1$, and $\hat{\mathcal{Z}}(i, j) = 0$ otherwise. For $\mathcal{L}_g$ and $\mathcal{L}_s$, we apply the same formulation, but we calculate the overlap ratio between two patches around the super points as ground-truth soft label values. This fact indicates that *we only provide a point-wise binary matrix for the supervision*, and the similarity matrix for super point matching and submap matching are calculated with $\hat{\mathcal{Z}}$. Additional supervision rather than the rigid transformation matrix between query and map point clouds is not required. More details on calculating ground-truth similarity matrices are provided in Appendix.

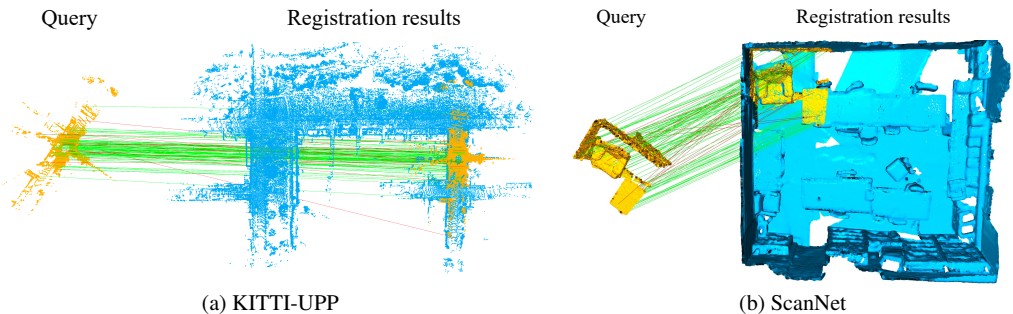

| Query | Registration results | Query | Registration results |

(a) KITTI-UPP      (b) ScanNet

Figure 2: Qualitative results on the KITTI-UPP dataset (a) and ScanNet dataset (b). Left figure for each dataset shows query point cloud and right one shows our registration result on unbalanced point cloud pairs. The green lines indicate inlier correspondences and the red lines indicate the outlier correspondences.

## 4 EXPERIMENT

### 4.1 EXPERIMENTAL SETTINGS

**KITTI-UPP dataset.** To the best of our knowledge, there are no available benchmarks and public datasets targeting the large-scale unbalanced point cloud registration task. To validate the effectiveness of our approach, we introduce KITTI-UPP, a carefully designed dataset for large-scale unbalanced point cloud registration tasks. We build KITTI-UPP by aggregating sequential LiDAR frames for each scene provided by the KITTI Odometry benchmark (Geiger et al., 2012). To control spatial and density imbalance factor between map and the query as defined in Section 3.1, we tune two parameters when selecting KITTI frames for aggregation: *range* and *hop*. The range determines how many frames we use to construct a map, indicating the map size. If the range value is set to 500, we aggregate 500 consecutive LiDAR frames to build a single map. The hop indicates the frame jump for the LiDAR frame aggregation. If we set the hop value as 10, every $10th$ frame is used for the aggregation. In this manner, the hop controls the density of the aggregated point cloud.

In our experiment, we set 300 and 10 as the default values for range and hop, respectively, to train UPPNet and baseline approaches. We then utilize KITTI-UPP scenes made with other hops and ranges that are unseen during the training to analyze the registration performance on the various spatial and density imbalance factors. Note that we ensure that any query is not included in the map. When we aggregate every 10th frames to construct a map, i.e., $\{10 \cdot i | 0 \leq i \leq \lceil \frac{\text{range}}{10} \rceil \}$, then we select the query frames from a index set $\{10 \cdot i + 5 \,| 0 \leq i \leq \lceil \frac{\text{range}}{10} \rceil \}$ to ensure that the same frame is not used in both a query and a map.

**ScanNet benchmark.** To validate the robustness of our proposed method in large-scale indoor environments, we use ScanNet (Dai et al., 2017) benchmark which contains 2M RGB-D scans of over 707 unique indoor scenes. To evaluate the methods under an unbalanced environment, we use the provided reconstructed mesh of ScanNet as a map and register a single RGB-D frame with the map, thus being able to omit the labor-heavy process for generating maps as in KITTI-UPP dataset construction. For the unbalanced point cloud registration task. An example pair is illustrated in Figure 2. Specifically, we use a subset of ScanNet which consists of 12/2/5 scenes for training/validation/testing splits respectively where each scene contains a varying number of scans ranging from 8 to 113.

**Evaluation metric.** We use the standard metric to assess the pairwise registration accuracy using *Rotational Error* (RE): $\arccos \frac{\text{Tr}(\mathbf{R}^T \hat{\mathbf{R}}) - 1}{2}$ and relative *Translational Error* (TE): $\|\mathbf{t} - \hat{\mathbf{t}}\|_2$, where $\mathbf{R}, \mathbf{t}$ are the predicted rotation matrix and a translation vector, $\hat{\mathbf{R}}, \hat{\mathbf{t}}$ are the ground-truth. We also use relative *Inlier Ratio* (IR) and *Registration Recall* (RR) for the evaluation. IR is defined as the ratio of correspondences whose geometric distances are below the predefined threshold ($\tau_I$) when transformed with the ground-truth transformation. For a registered pair having RE and TE less than the predefined thresholds ($\tau_{\mathbf{R}}, \tau_{\mathbf{t}}$), we regard it as a successful registration and calculate the RR of successful registration over the entire dataset. For indoor datasets, we report three standard metrics,

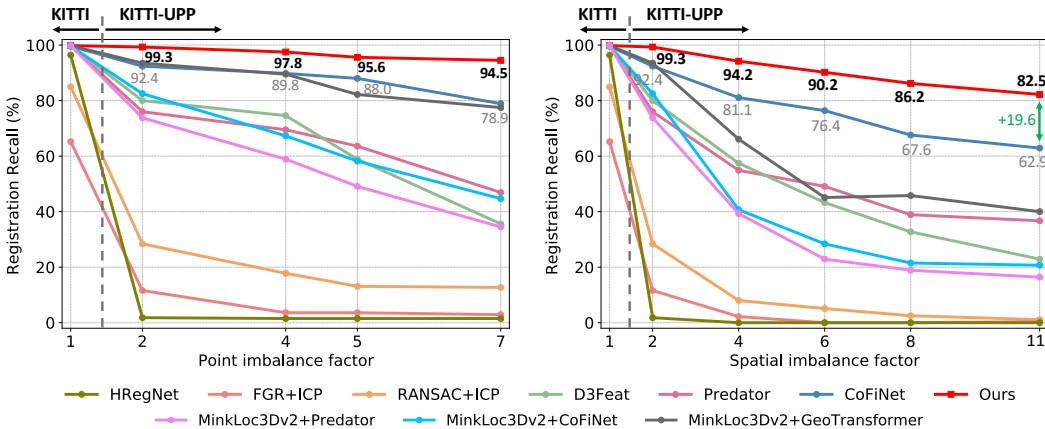

Figure 3: Evaluation results on KITTI and KITTI-UPP benchmark with various spatial ($\rho_s$) and point ($\rho_p$) imbalance factors. The regions on the left side of the gray lines indicate the balanced pairwise registration environment of the standard KITTI dataset on which the previous pairwise registration methods mainly handle. For the experiments with various point imbalance factors (left), we fix the range value to 100 ($\rho_s = 2.7$) and change the hop value. For the experiments with various spatial imbalances (right), we fix the hop value to 25 ($\rho_p = 6.9$) and change the range value.

Table 1: Quantitative registration results on 3DMatch (Zeng et al., 2017), 3DLoMatch (Huang et al., 2021), and ScanNet (Dai et al., 2017).

| | 3DMatch (Zeng et al., 2017) | | | 3DLoMatch (Huang et al., 2021) | | | ScanNet Dai et al. (2017) | | |
|---|---|---|---|---|---|---|---|---|---|
| | RR (%) | FMR (%) | IR (%) | RR (%) | FMR (%) | IR (%) | RR (%) | FMR (%) | IR (%) |
| FCGF (Choy et al., 2019b) | 85.1 | 97.4 | 56.8 | 40.1 | 76.6 | 21.4 | - | - | - |
| D3Feat (Bai et al., 2020) | 81.6 | 95.6 | 39.0 | 37.2 | 67.3 | 13.2 | - | - | - |
| Predator (Huang et al., 2021) | 89.0 | 96.6 | 58.0 | 59.8 | 78.6 | 26.7 | - | - | - |
| CoFiNet (Yu et al., 2021) | 89.3 | **98.1** | 49.8 | 67.5 | **83.1** | 26.7 | 50.0 | **84.2** | 51.0 |
| Ours | **93.0** | 96.8 | **72.4** | **69.0** | 82.3 | **39.7** | **71.0** | 80.1 | **64.1** |

namely *Registration Recall* (RR), *Inlier ratio* (IR) and *Feature Matching Recall* (FMR) following the literature. The precise definition of evaluation metrics can be found in the Appendix.

## 4.2 COMPARISON

**Outdoor Experiment.** We compare our UPPNet with the state-of-the-art pairwise registration methods (Bai et al., 2020; Huang et al., 2021; Yu et al., 2021) in Figure 3. We select those methods as baselines since they are equipped with keypoint detector (Bai et al., 2020), overlap prediction module (Huang et al., 2021), or coarse-to-fine registration module (Yu et al., 2021), which are likely to be favorable for the unbalanced point cloud registration. For a fair comparison, *we finetune all baseline methods on our KITTI-UPP dataset*, where the baselines were pretrained with KITTI odometry dataset (Geiger et al., 2012). All details of the training and evaluation setup for the baseline models can be found in Appendix. As shown in Figure 3, our method achieves the best results in all metrics. Moreover, our method achieves the best generalization ability w.r.t the various spatial extent and density imbalances as shown in Figure 3, indicating the advantages of our method. To verify the effectiveness of UPPNet, we conduct additional experiments; incorporating retrieval method with baseline models. In this experiment, we combine pairwise registration methods with MinkLoc3Dv2 (Komorowski, 2022) that shows best performance on 3D retrieval task. For fair comparison, We train the MinkLoc3Dv2 (Komorowski, 2022) on our KITTI-UPP dataset for 200 epochs. Even though pairwise registration methods benefit from MinkLoc3Dv2 (Komorowski, 2022), UPPNet still outperforms baseline models by large margins. We conclude that the method of generating global descriptor in MinkLoc3Dv2 (Komorowski, 2022) is not suitable for our challenging scenario which is under extreme imbalance in terms of spatial extent and point density; aggregating the features of all points rather than reliable points in contrast to UPPNet.

**Indoor Experiment.** We report unbalanced point cloud registration results on large-scale indoor environments, e.g., ScanNet (Dai et al., 2017). In this experiment, we compare our UPPNet with CoFiNet (Yu et al., 2021). We take two methods of ours and CoFiNet pretrained on 3DMatch (Zeng et al., 2017) and finetune on ScanNet. Registration results are summarized in Table 1; our method

Table 2: Ablation study on (Left) $k$-nearest-neighbors, (Right) submap proposal and structured matching modules. The range value and hop value are set to 500 ($\rho_s = 11.1$) and 25 ($\rho_p = 11.7$).

| #$k$ | Recall (%) | TE (m) | RE (°) | IR (%) |
|---|---|---|---|---|
| 1 | 65.5 | 0.691 | 1.613 | **25.1** |
| 2 | 75.3 | 0.675 | 1.599 | 24.1 |
| 4 | 78.5 | 0.668 | 1.477 | 22.0 |
| 8 | 81.5 | 0.602 | 1.379 | 19.7 |
| 16 | **82.5** | 0.612 | **1.314** | 16.9 |
| 32 | 81.5 | **0.602** | 1.327 | 14.0 |

| Submap | Structured | | Recall (%) |
|---|---|---|---|
| | super point | point | |
| ✗ | ✗ | ✗ | 70.2 |
| ✗ | ✓ | ✓ | 80.0 |
| ✓ | ✗ | ✗ | 74.5 |
| ✓ | ✗ | ✓ | 73.8 |
| ✓ | ✓ | ✗ | 76.4 |
| ✓ | ✓ | ✓ | **82.5** |

achieves 21.1% *Registration Recall* improvements over CoFiNet, which reveals the robustness of our method for indoor unbalanced point cloud registration as well.

## 4.3 ANALYSIS

To study the effectiveness of the proposed UPPNet, we conduct extensive ablation experiments and report the results in Table 2. In Table 2, we conduct an ablation study on the core design choices of our method: k-nearest-neighbor values and usage of submap proposal and structured matching modules. As shown in Table 2, $k = 16$ is the optimal configuration for considering all metrics. In addition, we found that both submap proposal and structured matching modules bring significant improvement in registration recall, where the best performance is achieved when all modules are enabled.

Moreover, we evaluate our method on three popular benchmarks, namely 3DMatch (Zeng et al., 2017), 3DLoMatch (Huang et al., 2021), and KITTI (Geiger et al., 2012), to demonstrate the applicability of the proposed method in the balanced but partially overlapping setup. We compare our method with the state-of-the-art feature-matching based registration methods (Bai et al., 2020; Choy et al., 2019b; Yu et al., 2021; Huang et al., 2021) on 3DMatch and 3DLoMatch benchmarks. For the KITTI

Table 3: Model runtime comparisons on KITTI-UPP dataset. Range and hop values are the same with Table 2.

| | Recall (%) | Time (s) |
|---|---|---|
| D3Feat (Bai et al., 2020) | 22.9 | 0.44 |
| Predator (Huang et al., 2021) | 36.7 | **0.19** |
| CoFiNet Yu et al. (2021) | 62.9 | 0.62 |
| Ours($k$=4) | 78.5 | 1.03 |
| Ours($k$=16) | **82.5** | 2.28 |

odometry benchmark, we select HRegNet (Lu et al., 2021) and CoFiNet (Yu et al., 2021) as baselines since they exhibit strong performance on KITTI benchmark (Geiger et al., 2012). For these experiments, we follow the training and evaluation settings of Yu et al. (2021). As reported in Table 1 and Figure 3, our method exhibits competitive results from the previous state-of-the-art registration methods on both indoor and outdoor datasets, even with the extremely low overlap scenario. This result suggests that our method is not specialized for the KITTI-UPP benchmark but also applies to the indoor and low overlap environment. For more details on training and evaluation for the 3DMatch dataset, please refer to the Appendix.

Finally, we measure the latency of our method and report the breakdown of elapsed times for each module on the KITTI-UPP benchmark in comparison with CoFiNet (Yu et al., 2021). As reported in Table 3, our method takes 0.4 seconds more than CoFiNet but improves registration recall by 15.6%.

## 5 CONCLUSION

In this paper, we presented a neural architecture for unbalanced point cloud registration with extreme spatial scale and point density discrepancy. We proposed a hierarchical framework that finds inlier correspondences effectively by gradually reducing search space to tackle this problem. Our proposed method can handle scale differences by finding subregions that are likely to be overlapped with the query point cloud and estimating the correspondences via a coarse-to-fine matching module to the selected subregions. Finally, structured matching is applied to prune the noisy correspondences further. Our method outperforms the state-of-the-art methods by a large margin in extensive experiments on the challenging dataset.

**Limitations.** We use RANSAC to get the final rigid transformation parameter with the estimated correspondences. Leveraging a differentiable model estimator, e.g., weighted Procrustes method (Choy et al., 2020a), would be an interesting future research direction.

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

# A  APPENDIX

In this supplementary material, we provide the detailed algorithm of the structured matching procedure in Section A.1, the description of how we form the data splits of our KITTI-UPP dataset for training and evaluation in Section A.2, additional details on the experiments in Section A.3, the additional quantitative results in Section A.4 with the implementation details of baseline methods, equations for the loss terms in Section A.5, details of architectural configuration in Section A.6, and finally qualitative results in Section A.7.

## A.1  STRUCTURED MATCHING

We provide additional information on our structured matching. As in Section3.4, the initial correspondence set is extracted from similarity matrix $\bar{\mathcal{Z}}$. Simple strategy is to consider an entry $(i, j)$ with high confidence score in $\bar{\mathcal{Z}}$ as a valid correspondence as in (Yu et al., 2021):

$$\mathcal{C} = \{(i,j)|\bar{\mathcal{Z}}(i,j) > \tau_{\mathcal{Z}}\} \tag{7}$$

However, this strategy is prone to missing inlier correspondences in our unbalanced setting; that is difficult to extract consistent features from the same region of the map and query as the inlier correspondences often have low confidence scores. An alternative solution is to select confident correspondences relatively for each point rather than using an absolute threshold value. We utilize K-nearest-neighbor (KNN) search in feature space and retrieve the top-k candidate correspondences for each point having the highest inlier confidence values:

$$\mathcal{C} = \{(i,j)|\bar{\mathcal{Z}}(i,j) > \text{Topk}(\bar{\mathcal{Z}}(i,:)\} \tag{8}$$

This approach can guarantee to select k candidates for each point even if their confidence scores are low. Although more inlier correspondences can be obtained by KNN search, selecting extra correspondences leads to a high number of outliers. To cope with these challenges, we combine spatial compatibility with KNN search for outlier rejection. We first check whether each correspondence satisfies spatial compatibility with others, as illustrated in Figure 4(b). Assume that we have $N$ correspondences $\{(x_n, y_n)\}_{n=1}^N$ where $x_n, y_n \in \mathbb{R}^3$ is a pair of 3D points, i.e., $n$-th correspondence. The spatial compatibility matrix $\mathbf{S} \in \mathbb{R}^{N \times N}$ encodes relative distances of the correspondences such that $\mathbf{S}_{i,j} = \mathbb{1}[|d(x_i, x_j) - d(y_i, y_j)| < \theta]$ where $\theta$ is distance threshold; score of 1 is assigned to $\mathbf{S}_{i,j}$ if a pair of correspondences $i, j$ are spatially consistent and scores of 0 is given otherwise. In this toy example, $\{c1, c3, c5, c7\}$ are inlier correspondences and $\{c2, c4, c6, c8\}$ are outlier correspondences. Although we can reject $c6$ by sampling the correspondences with at least one compatible correspondence, we still suffer from the outliers $\{c2, c4, c8\}$. Notably, a large-scale map would have numerous repetitive structures, making it challenging to distinguish hard negative correspondences $\{c2, c4, c8\}$ using only their associated features, i.e., through first order matching. Instead, we can select correspondences with high compatibility scores, as described in Section 3.4, by counting the number of compatible correspondences for each correspondence. Furthermore, we can calculate *compatibility coherence* between two correspondences by calculating how many correspondences are compatible with both of them. The calculation of compatibility coherence scores of some correspondence $i$ is formulated as similarities (dot-product) between spatial consistency of correspondence $i$ and those of other correspondences: $\mathbf{C}_i = \mathbf{S}\mathbf{S}_{i,:}^T \in \mathbb{R}^N$ which counts the number of spatially-consistent matches that correspondence $i$ has in common with others. In other words, compatibility coherence score of correspondences $i$ and $j$ amounts to *the number of spatially-consistent correspondences they have in common*. As an specific example, correspondences $c1$ and $c3$ in Figure 4(a) have *two common* spatially-consistent correspondences of $c5$ and $c7$ which makes $\mathbf{C}_{1,3} = \mathbf{C}_{3,1} = 2$ while correspondences $c6$ and $c8$ do not have any common spatially-consistent correspondences so $\mathbf{C}_{6,8} = \mathbf{C}_{8,6} = 0$. In our experiments, we use the compatibility coherence because it shows better performance in Table 4. We can get compatibility coherence by simply multiplying the compatibility matrix in Figure 4(b). Then, we compute the average coherence value per correspondence. Finally, thresholding these values with t indicates which correspondences to keep for the final correspondence set $\{c1, c3, c5, c7\}$. In our case, we set this threshold as the mean value of the average coherence values, as shown in Figure 4(c). As a result, by combining this strategy with KNN search, we firstly can obtain as many inliers as possible and efficiently filter out the outliers through structured matching. Experimentally, combining these two approaches is the key to solving the noisy correspondence problem caused by feature ambiguity common in unbalanced point pairs.

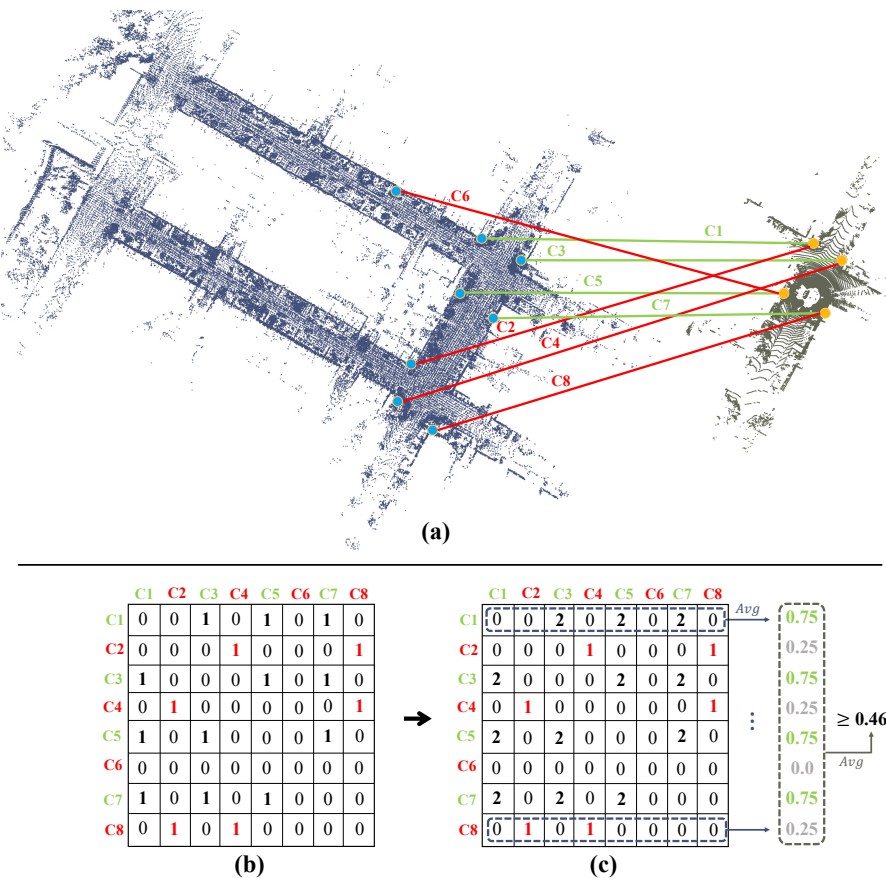

Figure 4: (a) Example of initial correspondences where green and red lines indicate inliers and outliers. (b) Spatial compatibility matrix. (c) Compatibility coherence matrix.

Table 4: Ablation study on compatibility matrix of score and similarity. The range value and hop value are set to 500 ($\rho_s = 11.1$) and 25 ($\rho_p = 11.7$).

| | Scale Imbalance Factor ($\rho_s$) : 11.1 | | | | |
| --- | --- | --- | --- | --- | --- |
| | Recall (%) ↑ | TE (m) ↓ | RE (°) ↓ | IR(%) ↑ | $\rho_P$ |
| Ours + compatibility score | 81.1 | **0.553** | 1.377 | 16.3 | 11.7 |
| Ours + compatibility coherence | **82.5** | 0.612 | **1.314** | **16.9** | |

## A.2 DATASET

We follow the same data split strategy that FCGF (Choy et al., 2019b) uses for KITTI odometry dataset (Geiger et al., 2012) for our KITTI-UPP dataset. We use sequences 0-5 for training, 6-7 for validation, and 8-10 for testing. For building the input point cloud pair, the query point clouds are sampled from the original frames of the KITTI odometry dataset, and the map point clouds are created by aggregating the LiDAR frames while managing two parameters: *range* and *hop* as described in Section 4.1. During training, we set the range and hop value to 300 and 10 to build maps. The query frames are at least 10m apart from each other, and we pick the map which is the closest one to each query among the generated map point clouds. For evaluation, we use varying range and hop values to evaluate the generalization ability of the methods. Especially for the test split, we carefully designed the input point cloud pairs so that the query would not be included in the frames that are used to generate the map. Through this procedure, we yield 1,358 pairs for training, 180 for validation, and 275 for testing.

### A.3  ADDITIONAL DETAILS ON THE EXPERIMENTS

**Evaluation metric.** We use the standard metric to assess the pairwise registration accuracy using *Rotational Error* (RE): $\arccos \frac{\mathrm{Tr}\,(\mathbf{R}^T \hat{\mathbf{R}}) - 1}{2}$ and relative *Translational Error* (TE): $\|\mathbf{t} - \hat{\mathbf{t}}\|_2$, where $\mathbf{R}, \mathbf{t}$ are the predicted rotation matrix and a translation vector, $\hat{\mathbf{R}}, \hat{\mathbf{t}}$ are the ground-truth. We also use relative *Inlier Ratio* (IR) and *Registration Recall* (RR) for the evaluation. IR is defined as the ratio of correspondences whose geometric distances are below the predefined threshold ($\tau_I$) when transformed with the ground-truth transformation. For a registered pair having RE and TE less than the predefined thresholds ($\tau_{\mathbf{R}}, \tau_{\mathbf{t}}$), we regard it as a successful registration and calculate the RR of successful registration over the entire dataset.

**Implementation details.** We implement our method with PyTorch (Paszke et al., 2019) and KP-Conv (Thomas et al., 2019) for efficient 3D kernel-point convolution. We use a U-shaped network equipped with an encoder and decoder, with three layers of KPConv and KPConv transposed layers. The feature dimensions of the point feature and super point feature are 32 and 256, respectively. Input point clouds are downsampled with a 1m voxel size on KITTI-UPP, 30cm on KITTI (Geiger et al., 2012), and 2.5cm on 3DMatch (Zeng et al., 2017) and 3DLoMatch (Huang et al., 2021). We set $\lambda_g = \lambda_p = 1$, and train the network for 200 epochs with Adam optimizer and initial learning rate of 1e-4. On inference time, $k$ is set to 16 for KNN search. The detailed architectural configuration can be found in  Figure 7. All experiments are performed on a single Nvidia Tesla V100 GPU and Intel Xeon Gold 6230R CPU @ 2.10GHz.

**Pairwise registration on 3DMatch and 3DLoMatch.** To evaluate our method on 3DMatch and 3DLoMatch dataset, we use the checkpoint of CoFiNet (Yu et al., 2021) pretrained on the 3DMatch dataset since the network architecture of our approach for the super point and point description is compatible with CoFiNet.

Following the recent literature (Choy et al., 2019b; Bai et al., 2020; Huang et al., 2021; Yu et al., 2021), we use three metrics to assess the registration performance of the methods:

- **Registration Recall (RR)**: The ratio of successful registration over the entire dataset. We consider the registration with RMSE ¡ 0.2 (m) as successful for 3DMatch and 3DLoMatch datasets.

- **Inlier Ratio (IR)**: The fraction of correct correspondences among the estimated correspondence set. We consider correspondences matched with smaller than 10cm residual as correct ones.

- **Feature Matching Recall (FMR)**: The fraction of point pairs with an inlier ratio higher than the predefined value over the entire dataset. We use 5% as the threshold for 3DMatch and 3DLoMatch datasets.

### A.4  ADDITIONAL QUANTITATIVE RESULTS

**More baselines.**    We provide the additional comparison of our UPPNet with the state-of-the-art pairwise registration methods (Bai et al., 2020; Huang et al., 2021; Yu et al., 2021; Lu et al., 2021) and classical methods (Zhou et al., 2016; Fischler & Bolles, 1981) in Table 5 and Table 6 with varying spatial and point imbalance factors. To compare with classical methods, we evaluate RANSAC (Fischler & Bolles, 1981) and FGR (Zhou et al., 2016) equipped with classical descriptor FPFH (Rusu et al., 2009) on our KITTI-UPP dataset. For RANSAC, we report the results with 4M iterations. They mostly fail on our challenging dataset. We further refine the registration results of both methods by applying post-processing with ICP (Besl & McKay, 1992). For state-of-the-art methods, we select Predator (Huang et al., 2021), CoFiNet (Yu et al., 2021), D3Feat (Bai et al., 2020), HRegNet (Lu et al., 2021) for the baseline methods as they are likely to be favorable for the unbalanced point cloud registration. For a fair comparison, we train the Predator, CoFiNet and ours for 200 epochs on our KITTI-UPP dataset. For D3Feat and HRegNet, the pretrained model trained on the KITTI Odometry dataset with 200 epochs is finetuned on our KITTI-UPP dataset with additional 50 epochs.

Table 5: Evaluation results on KITTI-UPP benchmark under various point imbalance factors ($\rho_p$) and scale imbalance factors ($\rho_s$). We evaluate the methods by changing hop value while we fix range value to (a) 100 ($\rho_s = 2.7$), (b) 200 ($\rho_s = 4.7$).

(a) Avg. Scale Imbalance Factor ($\rho_s$): 2.7

| Method | $\rho_p$ | RR (%)↑ | TE (m)↓ | RE (°)↓ | IR(%)↑ |
|---|---|---|---|---|---|
| FGR Zhou et al. (2016) | | 4.4 | 1.009 | 1.642 | 0.0 |
| FGR+ICP Park et al. (2017) | | 11.6 | 0.592 | 1.661 | 0.0 |
| RANSAC Fischler & Bolles (1981) | | 17.5 | 0.646 | 1.480 | 0.0 |
| RANSAC+ICP Park et al. (2017) | | 28.4 | 0.600 | 1.347 | 0.0 |
| HRegNet[†] Lu et al. (2021) | 2.3 | 5.1 | 0.900 | 1.456 | 1.5 |
| HRegNet Lu et al. (2021) | | 1.8 | 1.109 | 1.593 | 0.6 |
| D3Feat Bai et al. (2020) | | 80.0 | 0.609 | 1.252 | 8.9 |
| Predator Huang et al. (2021) | | 76.0 | 0.463 | 1.042 | 11.7 |
| CoFiNet Yu et al. (2021) | | 92.4 | 0.440 | 0.981 | 13.5 |
| Ours | | **99.3** | **0.359** | **0.762** | **20.9** |
| FGR Zhou et al. (2016) | | 1.8 | 0.741 | 1.607 | 0.0 |
| FGR+ICP Park et al. (2017) | | 3.6 | 0.545 | 2.212 | 0.0 |
| RANSAC Fischler & Bolles (1981) | | 4.7 | 1.047 | 1.659 | 0.0 |
| RANSAC+ICP Park et al. (2017) | | 17.8 | 0.804 | 1.537 | 0.0 |
| HRegNet[†] Lu et al. (2021) | 3.7 | 1.5 | 0.197 | 0.454 | 0.8 |
| HRegNet Lu et al. (2021) | | 1.5 | 1.073 | 2.341 | 0.4 |
| D3Feat Bai et al. (2020) | | 74.6 | 0.689 | 1.697 | 7.5 |
| Predator Huang et al. (2021) | | 69.5 | 0.545 | 1.221 | 8.8 |
| CoFiNet Yu et al. (2021) | | 88.4 | 0.551 | 1.289 | 13.1 |
| Ours | | **97.8** | **0.407** | **0.890** | **21.4** |
| FGR Zhou et al. (2016) | | 1.8 | 0.839 | 1.655 | 0.0 |
| FGR+ICP Park et al. (2017) | | 3.6 | 0.525 | 2.272 | 0.0 |
| RANSAC Fischler & Bolles (1981) | | 3.6 | 0.868 | 1.813 | 0.0 |
| RANSAC+ICP Park et al. (2017) | | 13.1 | 0.763 | 1.767 | 0.0 |
| HRegNet[†] Lu et al. (2021) | 4.8 | 1.8 | 0.569 | 0.690 | 0.7 |
| HRegNet Lu et al. (2021) | | 1.5 | 1.495 | 1.731 | 0.2 |
| D3Feat Bai et al. (2020) | | 58.9 | 0.747 | 1.802 | 6.3 |
| Predator Huang et al. (2021) | | 63.6 | 0.552 | 1.398 | 7.5 |
| CoFiNet Yu et al. (2021) | | 86.5 | 0.623 | 1.339 | 12.8 |
| Ours | | **95.6** | **0.412** | **0.902** | **21.7** |
| FGR Zhou et al. (2016) | | 1.1 | 1.287 | 1.598 | 0.0 |
| FGR+ICP Park et al. (2017) | | 2.9 | 0.596 | 2.349 | 0.0 |
| RANSAC Fischler & Bolles (1981) | | 1.5 | 1.307 | 1.688 | 0.0 |
| RANSAC+ICP Park et al. (2017) | | 12.7 | 0.714 | 1.918 | 0.0 |
| HRegNet[†] Lu et al. (2021) | 6.9 | 1.5 | 0.529 | 0.673 | 0.5 |
| HRegNet Lu et al. (2021) | | 0.0 | - | - | - |
| D3Feat Bai et al. (2020) | | 35.6 | 0.870 | 1.891 | 4.3 |
| Predator Huang et al. (2021) | | 46.9 | 0.694 | 1.600 | 5.2 |
| CoFiNet Yu et al. (2021) | | 76.0 | 0.706 | 1.486 | 11.3 |
| Ours | | **94.5** | **0.494** | **1.004** | **20.4** |

(b) Avg. Scale Imbalance Factor ($\rho_s$): 4.7

| Method | $\rho_p$ | RR (%)↑ | TE (m)↓ | RE (°)↓ | IR(%)↑ |
|---|---|---|---|---|---|
| FGR Zhou et al. (2016) | | 0.7 | 0.660 | 1.783 | 0.0 |
| FGR+ICP Park et al. (2017) | | 2.2 | 0.713 | 2.541 | 0.0 |
| RANSAC Fischler & Bolles (1981) | | 1.8 | 1.203 | 2.095 | 0.0 |
| RANSAC+ICP Park et al. (2017) | | 8.0 | 0.637 | 2.039 | 0.0 |
| HRegNet[†] Lu et al. (2021) | 4.5 | 0.7 | 0.223 | 1.335 | 0.2 |
| HRegNet Lu et al. (2021) | | 0.0 | - | - | - |
| D3Feat Bai et al. (2020) | | 57.5 | 0.753 | 1.717 | 5.4 |
| Predator Huang et al. (2021) | | 54.9 | 0.530 | 1.409 | 6.9 |
| CoFiNet Yu et al. (2021) | | 81.1 | 0.576 | 1.334 | 12.0 |
| Ours | | **94.2** | **0.445** | **1.042** | **20.6** |
| FGR Zhou et al. (2016) | | 0.4 | 0.829 | 1.759 | 0.0 |
| FGR+ICP Park et al. (2017) | | 0.7 | 0.552 | 2.350 | 0.0 |
| RANSAC Fischler & Bolles (1981) | | 1.1 | 1.100 | 2.035 | 0.0 |
| RANSAC+ICP Park et al. (2017) | | 4.0 | 0.872 | 2.006 | 0.0 |
| HRegNet[†] Lu et al. (2021) | 6.9 | 0.4 | 0.991 | 1.150 | 0.1 |
| HRegNet Lu et al. (2021) | | 0.4 | 1.971 | 3.969 | 0.1 |
| D3Feat Bai et al. (2020) | | 46.6 | 0.915 | 2.044 | 4.3 |
| Predator Huang et al. (2021) | | 56.4 | 0.631 | 1.518 | 5.4 |
| CoFiNet Yu et al. (2021) | | 77.5 | 0.671 | 1.673 | 12.0 |
| Ours | | **93.1** | **0.515** | **1.132** | **20.2** |
| FGR Zhou et al. (2016) | | 0.0 | - | - | - |
| FGR+ICP Park et al. (2017) | | 0.4 | 0.409 | 2.626 | 0.0 |
| RANSAC Fischler & Bolles (1981) | | 1.5 | 1.457 | 2.187 | 0.0 |
| RANSAC+ICP Park et al. (2017) | | 3.3 | 0.942 | 2.258 | 0.0 |
| HRegNet[†] Lu et al. (2021) | 9.2 | 0.4 | 1.186 | 1.618 | 0.1 |
| HRegNet Lu et al. (2021) | | 0.0 | - | - | - |
| D3Feat Bai et al. (2020) | | 31.3 | 0.926 | 2.152 | 3.5 |
| Predator Huang et al. (2021) | | 52.7 | 0.758 | 1.787 | 4.5 |
| CoFiNet Yu et al. (2021) | | 70.9 | 0.698 | 1.635 | 11.4 |
| Ours | | **90.5** | **0.515** | **1.105** | **19.2** |
| FGR Zhou et al. (2016) | | 0.0 | - | - | - |
| FGR+ICP Park et al. (2017) | | 0.7 | 0.728 | 2.659 | 0.0 |
| RANSAC Fischler & Bolles (1981) | | 0.4 | 1.572 | 1.996 | 0.0 |
| RANSAC+ICP Park et al. (2017) | | 3.3 | 1.002 | 2.167 | 0.0 |
| HRegNet[†] Lu et al. (2021) | 13.0 | 0.0 | - | - | - |
| HRegNet Lu et al. (2021) | | 0.0 | - | - | - |
| D3Feat Bai et al. (2020) | | 15.6 | 1.086 | 2.165 | 2.2 |
| Predator Huang et al. (2021) | | 33.8 | 0.727 | 1.919 | 3.0 |
| CoFiNet Yu et al. (2021) | | 56.0 | 0.806 | 1.721 | 9.6 |
| Ours | | **84.4** | **0.595** | **1.183** | **16.8** |

**More results with varying imbalance factors.** To evaluate the generalizability, we compare our UPPNet with the baseline methods under various scales and densities in Table 5 and Table 6. In Table 5a, we report the results with the fixed range value 100 under different hop value 25 ($\rho_p = 2.3$), 10 ($\rho_p = 3.7$), 5 ($\rho_p = 4.8$), and 1 ($\rho_p = 6.9$). Note that the range value 100 with hop value 25 is the least challenging data which is most similar to a balanced dataset e.g., KITTI odometry dataset. Likewise, the results in Table 5b, 6a, 6b are reported under various hop value 25, 10, 5, and 1 with fixed range value 200, 300, and 400, respectively. As shown in Table 5a-6b, performance of D3Feat (Bai et al., 2020) and CoFiNet (Yu et al., 2021) drops dramatically as the spatial extent and density of map become larger. On the other hand, our UPPNet can maintain the high performance at all evaluation metrics and outperform the baseline methods by a large margin, with robust structured matching and our hierarchical framework.

**Registration results with different overlap ratios.** To verify the robustness of our method with respect to the overlap ratio between query and map, we conduct experiments under varying overlap ratios as well as in presence of spatial and density imbalance. In this experiment, we set the range and hop values to 100 and 25 respectively. Following FCGF (Choy et al., 2019b), each pair is at least 10m apart. We compute overlap ratio for each pair and assign each pairs to three different groups; pairs with $< 70\%$, 70-85%, and $> 85\%$ overlap, and denote each group as *low, mid, and high* overlaps respectively. The results are shown in Figure 5 as well as the results of two baselines methods, CoFiNet (Yu et al., 2021) and Predator (Huang et al., 2021) As shown in the figure, our method consistently outperforms other baselines by large margin for all overlap ratios of low, mid, and high, gaining notable registration recall improvement of 26% over CoFiNet. We note that our method is not only effective in presence of spatial density imbalance but it is also robust under different overlap ratios.

Table 6: Evaluation results on KITTI-UPP benchmark under various point imbalance factors ($\rho_p$) and scale imbalance factors ($\rho_s$). We evaluate the methods by changing hop value while we fix range value to (a) 300 ($\rho_s = 6.8$), and (b) 400 ($\rho_s = 9.0$).

| | Avg. Scale Imbalance Factor ($\rho_s$) : 6.8 | | | | | | Avg. Scale Imbalance Factor ($\rho_s$) : 9.0 | | | |
|---|---|---|---|---|---|---|---|---|---|---|
| | $\rho_p$ | RR (%) ↑ | TE (m) ↓ | RE (°) ↓ | IR(%) ↑ | | $\rho_p$ | RR (%) ↑ | TE (m) ↓ | RE (°) ↓ | IR(%) ↑ |

| | $\rho_p$ | RR (%) ↑ | TE (m) ↓ | RE (°) ↓ | IR(%) ↑ | | $\rho_p$ | RR (%) ↑ | TE (m) ↓ | RE (°) ↓ | IR(%) ↑ |
|---|---|---|---|---|---|---|---|---|---|---|---|
| FGR Zhou et al. (2016) | | 0.0 | - | - | - | FGR Zhou et al. (2016) | | 0.0 | - | - | - |
| FGR+ICP Park et al. (2017) | | 0.0 | - | - | - | FGR+ICP Park et al. (2017) | | 0.0 | - | - | - |
| RANSAC Fischler & Bolles (1981) | | 0.7 | 1.425 | 1.714 | 0.0 | RANSAC Fischler & Bolles (1981) | | 0.4 | 1.102 | 1.835 | 0.0 |
| RANSAC+ICP Park et al. (2017) | | 5.1 | 0.920 | 2.255 | 0.0 | RANSAC+ICP Park et al. (2017) | | 2.5 | 0.830 | 2.296 | 0.0 |
| HRegNet[†] Lu et al. (2021) | 7.0 | 0.0 | - | - | - | HRegNet[†] Lu et al. (2021) | 8.9 | 0.0 | - | - | - |
| HRegNet Lu et al. (2021) | | 0.0 | - | - | - | HRegNet Lu et al. (2021) | | 0.0 | - | - | - |
| D3Feat Bai et al. (2020) | | 43.3 | 0.870 | 1.806 | 4.3 | D3Feat Bai et al. (2020) | | 32.7 | 0.990 | 2.023 | 3.3 |
| Predator Huang et al. (2021) | | 49.1 | 0.624 | 1.467 | 5.1 | Predator Huang et al. (2021) | | 38.9 | 0.686 | 1.732 | 3.8 |
| CoFiNet Yu et al. (2021) | | 76.4 | 0.716 | 1.555 | 11.6 | CoFiNet Yu et al. (2021) | | 67.6 | 0.737 | 1.747 | 11.3 |
| Ours | | **90.2** | **0.542** | **1.196** | **19.2** | Ours | | **86.2** | **0.579** | **1.327** | **18.5** |
| FGR Zhou et al. (2016) | | 0.0 | - | - | - | FGR Zhou et al. (2016) | | 0.0 | - | - | - |
| FGR+ICP Park et al. (2017) | | 0.0 | - | - | - | FGR+ICP Park et al. (2017) | | 0.0 | - | - | - |
| RANSAC Fischler & Bolles (1981) | | 0.7 | 1.470 | 1.921 | 0.0 | RANSAC Fischler & Bolles (1981) | | 0.7 | 1.298 | 1.689 | 0.0 |
| RANSAC+ICP Park et al. (2017) | | 2.5 | 0.640 | 2.262 | 0.0 | RANSAC+ICP Park et al. (2017) | | 2.5 | 0.931 | 2.336 | 0.0 |
| HRegNet[†] Lu et al. (2021) | 10.6 | 0.0 | - | - | - | HRegNet[†] Lu et al. (2021) | 13.5 | 0.0 | - | - | - |
| HRegNet Lu et al. (2021) | | 0.0 | - | - | - | HRegNet Lu et al. (2021) | | 0.0 | - | - | - |
| D3Feat Bai et al. (2020) | | 31.3 | 0.902 | 2.162 | 3.1 | D3Feat Bai et al. (2020) | | 19.6 | 1.001 | 2.148 | 2.6 |
| Predator Huang et al. (2021) | | 45.5 | 0.721 | 1.859 | 4.1 | Predator Huang et al. (2021) | | 34.5 | 0.764 | 1.841 | 3.1 |
| CoFiNet Yu et al. (2021) | | 64.7 | 0.834 | 1.818 | 11.1 | CoFiNet Yu et al. (2021) | | 58.9 | 0.782 | 1.990 | 10.3 |
| Ours | | **85.1** | **0.592** | **1.375** | **18.0** | Ours | | **79.6** | **0.608** | **1.291** | **16.8** |
| FGR Zhou et al. (2016) | | 0.0 | - | - | - | FGR Zhou et al. (2016) | | 0.0 | - | - | - |
| FGR+ICP Park et al. (2017) | | 0.0 | - | - | - | FGR+ICP Park et al. (2017) | | 0.0 | - | - | - |
| RANSAC Fischler & Bolles (1981) | | 0.4 | 1.424 | 2.007 | 0.0 | RANSAC Fischler & Bolles (1981) | | 0.4 | 1.692 | 1.925 | 0.0 |
| RANSAC+ICP Park et al. (2017) | | 2.9 | 0.660 | 2.213 | 0.0 | RANSAC+ICP Park et al. (2017) | | 1.5 | 0.738 | 2.212 | 0.0 |
| HRegNet[†] Lu et al. (2021) | 13.4 | 0.0 | - | - | - | HRegNet[†] Lu et al. (2021) | 18.8 | 0.0 | - | - | - |
| HRegNet Lu et al. (2021) | | 0.0 | - | - | - | HRegNet Lu et al. (2021) | | 0.0 | - | - | - |
| D3Feat Bai et al. (2020) | | 21.8 | 0.875 | 2.119 | 2.4 | D3Feat Bai et al. (2020) | | 13.5 | 1.127 | 2.056 | 1.8 |
| Predator Huang et al. (2021) | | 35.6 | 0.687 | 1.787 | 3.2 | Predator Huang et al. (2021) | | 25.1 | 0.786 | 1.938 | 2.2 |
| CoFiNet Yu et al. (2021) | | 61.1 | 0.795 | 1.895 | 10.4 | CoFiNet Yu et al. (2021) | | 52.4 | 0.837 | 1.929 | 9.0 |
| Ours | | **81.8** | **0.631** | **1.266** | **16.7** | Ours | | **78.5** | **0.617** | **1.333** | **15.2** |
| FGR Zhou et al. (2016) | | 0.0 | - | - | - | FGR Zhou et al. (2016) | | 0.0 | - | - | - |
| FGR+ICP Park et al. (2017) | | 0.0 | - | - | - | FGR+ICP Park et al. (2017) | | 0.0 | - | - | - |
| RANSAC Fischler & Bolles (1981) | | 0.0 | - | - | - | RANSAC Fischler & Bolles (1981) | | 0.0 | - | - | - |
| RANSAC+ICP Park et al. (2017) | | 1.5 | 1.072 | 2.433 | 0.0 | RANSAC+ICP Park et al. (2017) | | 0.7 | 1.092 | 2.400 | 0.0 |
| HRegNet[†] Lu et al. (2021) | 18.8 | 0.0 | - | - | - | HRegNet[†] Lu et al. (2021) | 26.3 | 0.0 | - | - | - |
| HRegNet Lu et al. (2021) | | 0.0 | - | - | - | HRegNet Lu et al. (2021) | | 0.0 | - | - | - |
| D3Feat Bai et al. (2020) | | 10.9 | 1.050 | 1.749 | 1.5 | D3Feat Bai et al. (2020) | | 7.3 | 1.137 | 1.854 | 1.1 |
| Predator Huang et al. (2021) | | 18.5 | 0.792 | 1.939 | 2.0 | Predator Huang et al. (2021) | | 13.5 | 0.929 | 2.128 | 1.3 |
| CoFiNet Yu et al. (2021) | | 48.0 | 0.869 | 2.121 | 8.2 | CoFiNet Yu et al. (2021) | | 42.5 | 0.813 | 2.029 | 6.8 |
| Ours | | **70.9** | **0.651** | **1.256** | **13.8** | Ours | | **71.6** | **0.674** | **1.460** | **12.9** |

(a)                                                            (b)

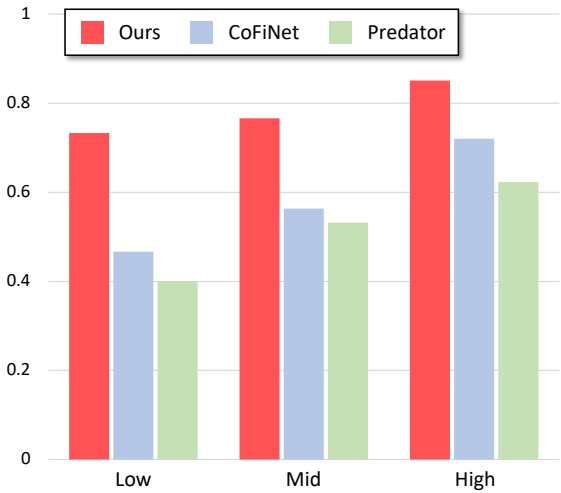

Figure 5: Evaluation results on KITTI-UPP benchmark with different overlap ratios. For the experiments with different overlap ratios, we fix the range value to 100 ($\rho_s = 2.3$) and the hop value to 25 ($\rho_p = 2.3$).

**Registration results with ground-truth submap.** To validate the effectiveness of our proposed submap matching and structured matching modules, we analyze the registration performance of all methods when the ground-truth submap is provided. Note that this setting is advantageous for the

baseline methods since they are optimized for balanced point pairs, and we refer to these results as the theoretical upperbounds of the baselines methods since we can assume the combination of a highly accurate localization algorithm with the baseline methods. As shown in Figure 6, thanks to the robustness of the submap matching module in establishing reliable submap correspondences, our model without gt submaps achieves relatively small performance drops compared to others (Yu et al., 2021; Huang et al., 2021). Moreover, we highlight that the proposed structured matching effectively helps our method achieve the best results, outperforming the upper bounds of other baselines by large margins even without ground truth submaps.

## A.5 LOSS

In this section, we provide concise equations to calculate the loss terms, $\mathcal{L}_s$, $\mathcal{L}_g$, and $\mathcal{L}_p$. As we described in Section 3.6 in our main manuscript, the point matching loss $\mathcal{L}_p$ is defined as:

$$\mathcal{L}_{\mathrm{p}} = \frac{-\sum_{i,j} \hat{\mathcal{Z}}(i,j) \log \bar{\mathcal{Z}}(i,j)}{\sum_{i,j} \hat{\mathcal{Z}}(i,j)}, \tag{9}$$

where $\bar{\mathcal{Z}}$ is the predicted similarity matrix after solving optimal transport using the Sinkhorn algorithm, $\hat{\mathcal{Z}}$ is the binary matrix that indicates ground-truth. For coarse scale super point and submap level matching, we follow (Yu et al., 2021) and calculate the overlap ratio between two super points, or submaps, to calculate soft-labeled supervision. For super point matching, we calculate the overlap ratio between two super points as:

$$r(i', j') = \frac{|\{\mathbf{x} \in \mathbf{G}_{i'}^{\mathbf{X}} | \exists \mathbf{y} \in \mathbf{G}_{j'}^{\mathbf{Y}}, \mathrm{s.t.} \|\hat{\mathbf{R}}\mathbf{y} + \hat{\mathbf{t}} - \mathbf{x}\| < \tau_g|}{|\mathbf{G}_{i'}|} \tag{10}$$

where $\hat{\mathbf{R}}, \hat{\mathbf{t}}$ is the groundtruth rotation and translation and $\tau_g$ is the predefined distance threshold. If the superpoint $(i', j')$ is not an inlier correspondence, we assign it to the slack entry in the similarity matrix. For this, we calculate the ratio of points in $\mathbf{G}_{i'}$ which are overlapped with point cloud $\mathbf{Y}$:

$$r(i') = \frac{|\{\mathbf{x} \in \mathbf{G}_{i'}^{\mathbf{X}} | \exists \mathbf{y} \in \mathbf{Y}, \mathrm{s.t.} \|\hat{\mathbf{R}}\mathbf{y} + \hat{\mathbf{t}} - \mathbf{x}\| < \tau_g|}{|\mathbf{G}_{i'}|} \tag{11}$$

and finally build the groundtruth similarity matrix $\hat{\mathcal{Z}}_g \in \mathbb{R}^{n'+1 \times m'+1}$ as:

$$\hat{\mathcal{Z}}_g(i', j') = \begin{cases} \min(r(i', j'), r(j', i')) & \text{if } i' < n' + 1 \wedge j' < m' + 1 \\ 1 - r(i') & \text{if } i' < n' + 1 \wedge j' = m' + 1 \\ 1 - r(j') & \text{if } i' = n' + 1 \wedge j' < m' + 1 \\ 0 & \text{otherwise} \end{cases} \tag{12}$$

Then we calculate the loss similar with point matching loss.

$$\mathcal{L}_g = \frac{-\sum_{i',j'} \hat{\mathcal{Z}}_g(i', j') \log \bar{\mathcal{Z}}_g(i', j')}{\sum_{i',j'} \hat{\mathcal{Z}}_g(i', j')} \tag{13}$$

Note that unlike the point matching loss $\mathcal{L}_p$, super point matching loss is supervised with the continuous soft labeled supervision $\hat{\mathcal{Z}}_g$. Similarly, we can calculate the submap matching loss $\mathcal{L}_s$:

$$\mathcal{L}_s = \frac{-\sum_{i',j'} \hat{\mathcal{Z}}_s(i', j') \log \bar{\mathcal{Z}}_s(i', j')}{\sum_{i',j'} \hat{\mathcal{Z}}_s(i', j')}, \tag{14}$$

where $\hat{\mathcal{Z}}_s$ is the ground-truth similarity matrix at submap level, and it can be calculated in a similar way.

## A.6 NETWORK ARCHITECTURE

UPPNet adopt a shared U-shaped network based on KPConv (Thomas et al., 2019) to extract multi-level features. Detailed architecture is demonstrated in Figure 7. Compared to CoFiNet (Yu et al., 2021), additional layers for the submap matching module are added to our network. We generate a global descriptor through Generalized Mean Pooling (GeM). As the global descriptor of the query is a single vector, the self-attention module is applied only for global descriptors of the map, such as in Figure 7.

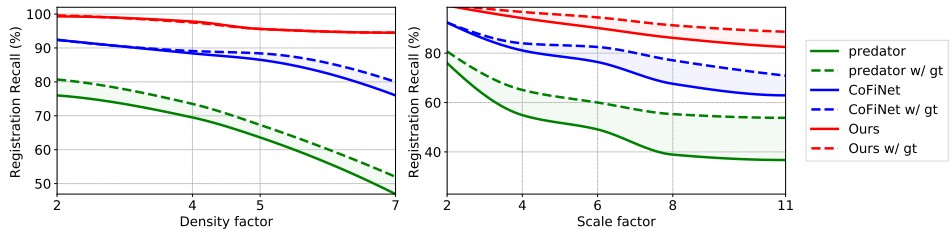

Figure 6: *Registration Recall* in relation to (**Left**) point imbalance factor ($\rho_p$) and (**Right**) scale imbalance factor ($\rho_s$). The dotted lines indicate the theoretical upper bounds of each method measured by providing a ground truth submap. Our method shows the best registration recall in all settings, and the margin to the upper bound is smaller than the baselines.

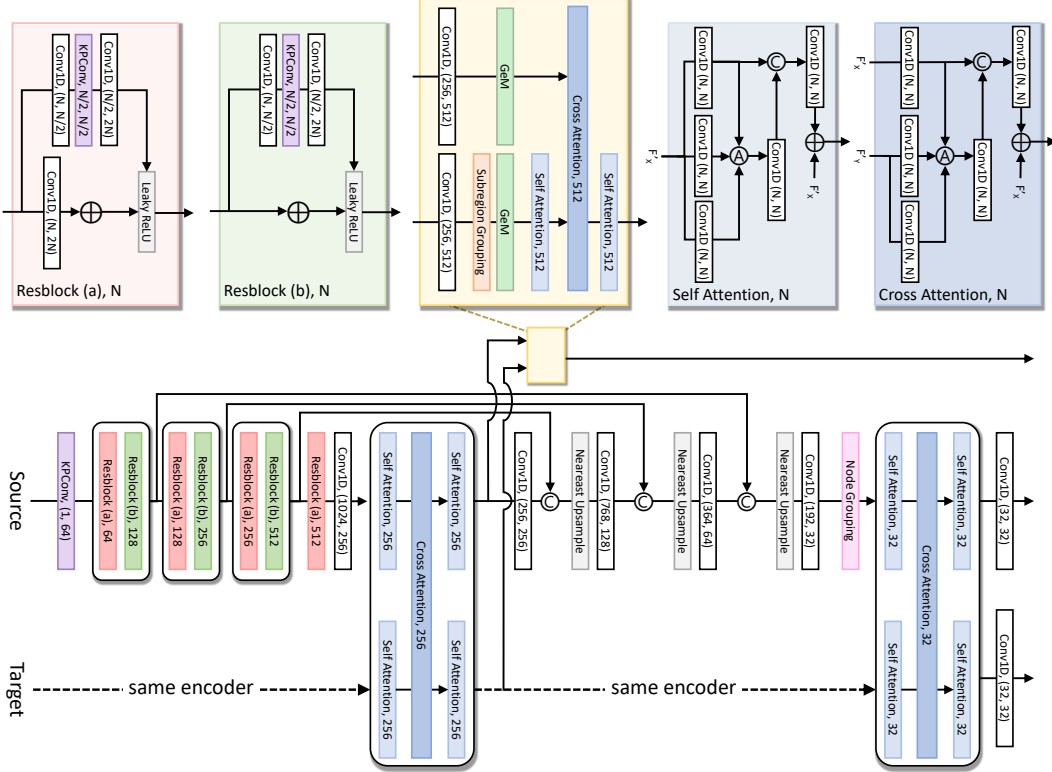

Figure 7: Network architecture of UPPNet.

## A.7    ADDITIONAL QUALITATIVE RESULTS

In Figure 8, we provide additional qualitative results of our method along with the baseline methods on the KITTI-UPP test dataset.

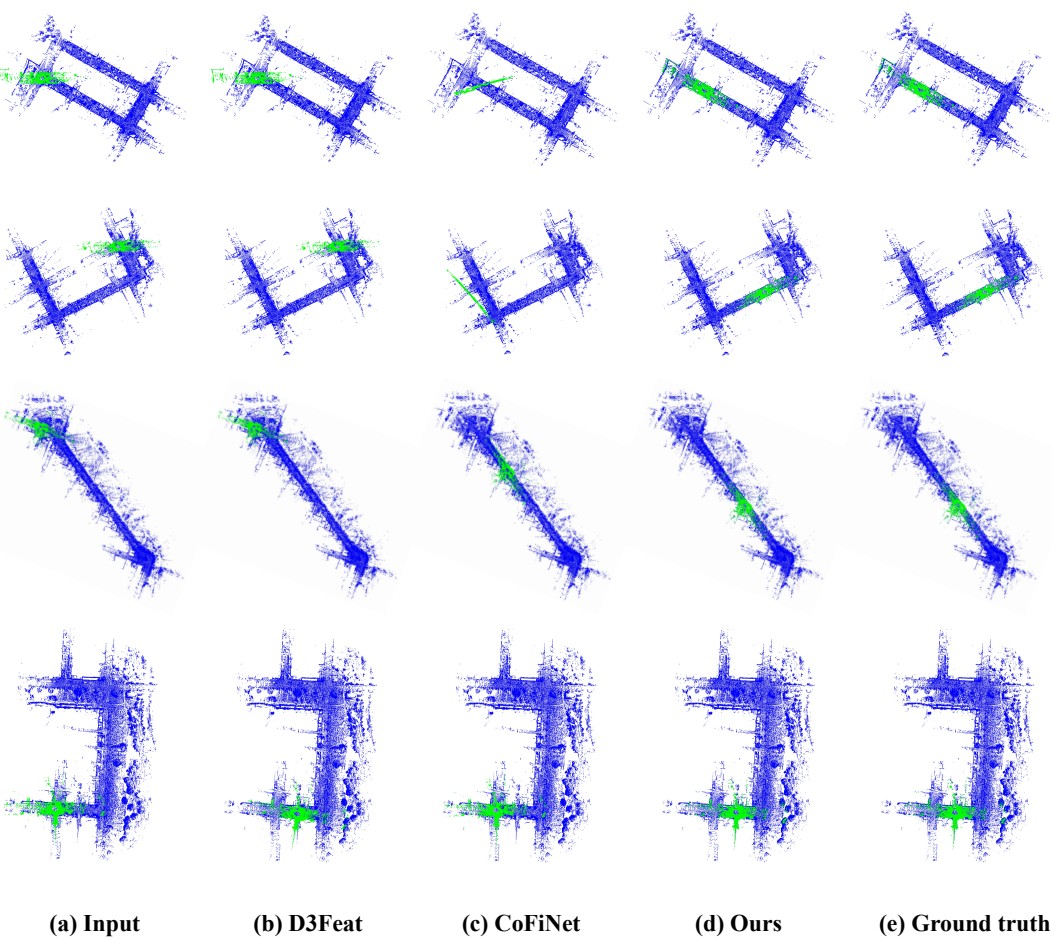

(a) Input      (b) D3Feat      (c) CoFiNet      (d) Ours      (e) Ground truth

Figure 8: Registration results on KITTI-UPP dataset. (Green) A query. (Blue) A map.

