# OpenReview forum: "Learning to Register Unbalanced Point Pairs"
_ICLR.cc/2023/Conference — Submitted to ICLR 2023_

### Official Review · Reviewer_CTpR · 2022-10-24

**Confidence:** 5
**Correctness:** 3
**Technical Novelty And Significance:** 2
**Empirical Novelty And Significance:** Not applicable
**Recommendation:** 6

**Clarity, Quality, Novelty And Reproducibility:**

[Clarity]: Good;
[Quality]: Good;
[Novelty]: Neutral;
[Reproducibility]: Good;

**Strength And Weaknesses:**

[Strength]

1. This paper studies a new registration problem;
2. This paper is well-written and easy to follow;
3. Experiment results are good;

[Weaknesses]

1. [One feasible solution..., above section 3].
Authors say that it is suboptimal to divide the proposed task into two subtasks : 1) the nearest frame retrieval and 2) local registration.  However, authors still follow this pipeline by first performing KNN submap retrieval and then do local registration.
The difference could be authors use an attention-based context aggregation module to extract better global features.
According to the above analysis, I would expect the following three experiments:
1) Substitute the proposed submap retrieval [Sec.3.3] with a state-of-the-art nearest frame retrieval method [e.g., https://github.com/jac99/MinkLoc3Dv2], and keep the remaining modules to test the proposed method;
2) Use a state-of-the-art nearest frame retrieval method for state-of-the-art methods to find a local submap for a query, and then test the performance of state-of-the-art methods;
3) Use the proposed submap retrieval [Sec.3.3]  for baseline methods to find a local submap for a query, and then test the performance of state-of-the-art methods;

The above three experiments (ablations) can help readers get a clear picture of the proposed method and identifying what is actually working.

2. It is blurry how the proposed method is tested on the standard 3DMatch and original KITTI datasets. Are you using K=1 and skip the submap matching module?

3. For comparison with respect to state-of-the-art methods (Table 1), please report the original figures of these methods. For example,
 [Predator, RR] 90.6 for 3DMatch and 62.4 for 3DLoMatch.
Furthermore, please also compare the state-of-the-art method [Geometric Transformer for Fast and Robust Point Cloud Registration], it got 92.5 for 3DMatch and 74.2 for 3DLoMatch.





**Summary Of The Paper:**

This paper defines a new task: matching unbalanced points cloud pairs in terms of spatial extent and density.

Though the task is somewhat similar with registering low-overlap point cloud pairs for spatial extent, this paper adds a factor - density.

This paper proposes three steps/modules to solve the problem: 1) Submap partition and matching to find top-K submaps of the large database for the query; 2)  coarse-level super-point matching; and 3) fine-grained point matching.

Results on the KITTI-UPP and several standard benchmark datasets show the effectiveness of the proposed method.




**Summary Of The Review:**

Though this paper proposes to solve a new task, some important baseline experiments are missing [please refer to weaknesses], this brings a question -- can we combine the best practice of a state-of-the-art nearest frame retrieval method [e.g., https://github.com/jac99/MinkLoc3Dv2] and a state-of-the-art registration method [Geometric Transformer for Fast and Robust Point Cloud Registration] to solve the proposed task?

---

> ### Author Response · Authors · 2022-11-25
> **Our response**
>
> > Q1: Authors say that it is suboptimal to divide the proposed task into two subtasks. However, authors still follow this pipeline.
>
> A: As we stated in the “General Comment Q1”, we clarified the motivation of our work and the difference with the Retrieval-then-Registration pipeline. Contrary to previous retrieval [7] and registration methods [1, 2, 3, 4, 5, 6], the proposed model is able to learn strong features for the submaps and local geometries under spatial extent and point density imbalance, thus being able to solve the conventional **two-stage** problems with a **single-stage** approach. Please refer to “General Comment Q1” for more details.
>
> > Q2: According to the above analysis, I would expect the following three experiments.
>
> A: As we stated in the “General Comment Q2”, we conducted additional experiments by combining the state-of-the-art point cloud retrieval method, MinkLoc3Dv2 [7], with the state-of-the-art pairwise registration methods [2,3,6]. The results show that our method outperforms the Retrieval-then-Registration pipeline by a large margin. Please refer to “General Comment Q2” for more detailed results and analysis.
>
> > Q3: It is blurry how the proposed method is tested on the standard 3DMatch and original KITTI datasets. Are you using K=1 and skip the submap matching module?
>
> A:
> Thank you for your time and helpful feedback. Our approach measures the ratio between the spatial extents of the two point clouds, and it skips the submap matching module when the target point cloud is not significantly larger than the source point cloud. However, our structured matching module still works to reject outliers that violate spatial consistency; the reason why our UPPNet could outperform baseline methods without the submap matching module on standard datasets as in Table 1 of the main paper.
>
>
> > Q4: For comparison with respect to state-of-the-art methods (Table 1), please report the original figures of these methods.
>
> A: Yes, we already did it. To be accurate,  the numbers for the baseline methods [2,3] on 3DMatch, 3DLoMatch, and KITTI in Table 1 of our main paper, we brought the results from [3]. The results are also reported in Figure 3 at the leftmost x-axis data points where the imbalance factors are 1.

---

> > ### Author Response · Authors · 2022-12-12
> > **Response to Reviewer CTpR**
> >
> > Dear respectful reviewer, we have added our responses in “General Comment Q1 and Q2”. We hope that our additional experiments and responses have addressed the reviewer’s concerns related to the motivation of our work and the difference between Retrieval-then-Registration pipelines.

---

### Official Review · Reviewer_fyXh · 2022-10-25

**Confidence:** 3
**Correctness:** 2
**Technical Novelty And Significance:** 3
**Empirical Novelty And Significance:** Not applicable
**Recommendation:** 5

**Clarity, Quality, Novelty And Reproducibility:**

One of the main issues I have with the paper is that the writing can be improved. Many parts are hard to follow or missing important details:
- For example, How is S_i determined exactly? Here, v is learnable, and yet the intersection between different submaps is constrained to be 0.5. How is this exactly achieved?
- In Appendix, the calculation of compatibility coherence is only briefly explained. Here, it says that the coherence is calculated by multiplying the compatibility matrix, but multiplying it with what?
- In Section 3.3., global features are calculated based on generalized mean pooling. Does this mean that F'_{x'_j}(k) is positive?

The proposed method is designed with many different small components, but the details are not 100% clearly explained in the paper. Therefore, I believe more information (such as the source code) is needed to reproduce the results.

**Strength And Weaknesses:**

The main motivation of the paper, unbalanced point cloud pairs, is a plausible problem setting that can be important for practical scenarios. Although many corners of the proposed method are inspired by existing techniques in the literature, they are combined in a way that the above problem can be solved effectively. The proposed method provides good performance in the experiment which is also a good point.

However, I also think that the proposed method has a limitation in its design. It seems that Y (the query) is assumed to be almost a subset of X. Unlike X, Y has only a single submap, which is matched with various submaps of X, as if we are searching for a part of X that corresponds to Y. However, a more general situation is when X and Y are only partially overlapped, and a large part of Y is also absent in X. Of course, I see that the authors are considering unmatched super points for both directions in (6), but I still somewhat doubt the choice of having a single submap for Y. It seems that this design simplifies the problem too much. I'd like to hear the authors' justification regarding this part.



**Summary Of The Paper:**

This paper proposes a new point cloud registration method considering an unbalanced pair of point clouds. The main assumption is that one of them contains much fewer points than the other and represents only a part of the scene in interest. There are several spatial levels as well as the corresponding matching modules in the proposed method. The points are downsampled to super points, which are then again grouped into several submaps. The point and super-point features are calculated by a KPConv-based network, and they are aggregated using attention to calculate those for submaps. These features are used in a hierarchical matching process comprised of submap matching, super point matching, and finally point matching. An additional RANSAC procedure is applied to the point correspondences to refine the results. Experiments show that the proposed method beats the existing methods by a large margin for a new dataset created considering the particular problem setting proposed in the paper, and it is competitive for existing datasets.

**Summary Of The Review:**

This paper proposes an important problem that often arises in practical scenarios, and achieves good performance in the experiment. However, I also have a doubt about its method design (regarding the treatment of Y). Moreover, many important details are missing or confusingly described in the paper.

[After rebuttal] I'm satisfied with the answers regarding my original concerns. However, I agree with g5jK regarding the fairness of the experiments, which is a non-negligible concern. Accordingly, I decrease my score to 5.

---

> ### Author Response · Authors · 2022-11-26
> **Our response**
>
> > Q1: More general situation is when X(Map) and Y(Query) are only partially overlapped, and a large part of Y is also absent in X. However, Y has only a single submap.
>
> A:
> Thanks for the constructive comments. In fact, the proposed method **does not assume the situation explicitly where the query point cloud (Y) is fully overlapped by the map (X)**. We have analyzed the robustness of our method on a partially overlapped environment, as shown in Figure 5 of supplementary. Unlike previous retrieval methods that generate global descriptors by aggregating the features of all points, we use the attention module to aggregate the features of important key points; our attention-based context aggregation module interactively aggregates global contexts across two point clouds. Thanks to these advantages of the attention module, our method outperforms baseline methods [2,3] in partially overlapped cases, even with the design choice of a single submap for Y. However, dividing the query Y into multiple subregions would be a generalized version of ours, and we think it is an interesting direction for future research.
>
> > Q2: How is $S_i$ determined exactly.
>
> A:
> A submap $\mathbf{S}\_{i}$ is a subset of superpoints of the map, i.e., $\mathbf{S}\_{i} \subset \mathbf{X}'$, that lie in $i$-th submap region $\mathcal{S}\_{i}$: $\mathbf{S}\_{i} = ${$\mathbf{x}' \in \mathbf{X}': \mathbf{x}' \in \mathcal{S}\_{i}$}.
> Each (cubic-shaped) submap region $\mathcal{S}\_{i}$ with its center $\mathbf{c}\_{i} \in \mathbb{R}^{3}$ (which we call center point) and edge length of $v \in \mathbb{R}^{+}$ is formally defined as $\mathcal{S}\_{i} = ${$\mathbf{s} \in \mathbb{R}^{3}: \text{max}|\mathbf{s} - \mathbf{c}\_i| \le v/2$}.
> In this work, we evenly place $L$ overlapping submap regions {$\mathcal{S}\_{i}$}$\_{i=1}^{L}$ to cover all the superpoints of the map $\mathbf{X}'$ with carefully chosen overlapping ratio $\mu$, satisfying $|\mathcal{S}\_{i} \cap \mathcal{S}\_{j}| / |\mathcal{S}\_{i}| = \mu$ and $||\mathbf{c}\_{i}-\mathbf{c}\_{j}|| = v \cdot \mu$ for any adjacent, overlapping submap regions $\mathcal{S}\_{i}$ and $\mathcal{S}\_{j}$.
> We set $\mu = 0.5$ in our experiments.
> The edge length of the submap regions, $v \in \mathbb{R}^{+}$, is defined by the spatial size of the query $\mathbf{Y}$.
> Specifically, we first compute the furthest distance between two points in $\mathbf{Y}$ along $i$-th axis as $s\_{i} = | \text{max}\_{a} \mathbf{y}\_{a, i} - \text{min}\_{b} \mathbf{y}\_{b, i} | \in \mathbb{R}^{+}$, thus giving {$s\_{i}$}$\_{i=1}^{3}$ of which respective elements represent width ($s\_1$), height ($s\_2$), and length ($s\_3$) of $\mathbf{Y}$.
> The edge length of submap regions is defined as the largest edge length of $\mathbf{Y}$: $v = \text{max}\_{i} s\_{i}$. **We clarify this in our revised version of the paper.**
>
> > Q3: Calculation of compatibility coherence is only briefly explained.
>
> A :
> Assume we have $N$ correspondences {$(x_n, y_n)$}$\_{n=1}^{N} $ where $x_n, y_n \in \mathbb{R}^{3}$ is a pair of 3D points, i.e., $n$-th correspondence. The spatial compatibility matrix $\mathbf{S}^{N \times N}$ (Fig. 4 (b) of the supp.) encodes relative distances of the correspondences such that $\mathbf{S}\_{i,j} = 1[ |d(x_i, x_j) – d(y_i, y_j)| < \theta]$ where $\theta$ is distance threshold; score of 1 is assigned to $\mathbf{S}\_{i, j}$ if a pair of correspondences $i, j$ are spatially consistent and scores of 0 is given otherwise. We set $\theta = 8\cdot$voxel size in our experiments. The calculation of compatibility coherence scores of some correspondence $i$ is formulated as similarities (dot-product) between spatial consistency of correspondence $i$ and those of other correspondences: $\mathbf{C}\_{i} = \mathbf{S}\mathbf{S}_i^T \in \mathbb{R}^{N}$ which **counts** the number of spatially-consistent matches that correspondence $i$ has in common with others. In other words, the compatibility coherence score of correspondences $i$ and $j$ amounts to *the number of spatially-consistent correspondences they have in common*. As an specific example, correspondences C1 and C3 in Fig. 4 (a) of supplementary have *two common* spatially-consistent correspondences of C5 and C7 which makes $\mathbf{C}\_{1, 3} = \mathbf{C}\_{3, 1} = 2$ while correspondences C6 and C8 do not have any common spatially-consistent correspondences so $\mathbf{C}\_{6, 8} = \mathbf{C}\_{8, 6} = 0$. **We clarify this in our revised version of the paper. Please refer to Appendix A.1 for more details.**
>
>
> > Q4: In Section 3.3., global features are calculated based on generalized mean pooling. Does this mean that F'_{x'_j}(k) is positive?
>
> A: Thanks for pointing out this issue. We leave out the implementation detail in this part. We clip the negative values of superpoint features F'_{x'_j} to 0 before applying generalized mean pooling to avoid a numerical error. Sorry for the confusion.

---

> > ### Comment · Reviewer_fyXh · 2022-12-03
> > **Thank you for the detailed answers and updates.**
> >
> > I'm satisfied with the answers for Q2-Q4. However, I still have some doubts about that for Q1. My original question was not intended for "fully overlapped" situation. Even though there are only partial overlaps, Y still can have a large area that cannot be briefly summarized using a single submap (and this can be "unoverlapped" parts). I see that other reviewers, too, have problems with the problem definition (even though the direction of criticism is somewhat different). I'm thinking of reducing my original score, and I'll decide my final score after the discussion reaches a consensus.

---

> > > ### Author Response · Authors · 2022-12-12
> > > **Our response**
> > >
> > > We politely request the reviewer to visit page 17 in the supplementary. We conducted experiments with the partially overlapped point cloud pairs in the original submission. We classified the cases of overlap depending on the low overlap (less than 70%), medium overlap (70-85%), and high overlap (more than 85%). In this experiment, ours outperforms other methods even with a single submap of a query, while CoFiNet and Predator fail under partially overlapped settings. For the partially overlapped settings, the distribution of the overlap ratio is shown in Figure 1. of the attached link[1].
> > >
> > > Regarding the problem settings, we have stated the problems of the retrieval & registration approaches that are another way to achieve unbalanced point pair registration. The Retrieval-then-Registration methods are two staged approaches that are suboptimal to learning robust geometric features. Also, They require several feed-forwards at the test time, and the retrieval module bounds the overall accuracy. We provide additional experimental results and show that the proposed approach outperforms the best combination of retrieval (MinkLoc3Dv2) and registration (Predator, CoFiNet, GeoTransformer) modules.
> > >
> > > We consider the imbalances based on the spatial and point density of two point clouds and capture imbalances between the point clouds. We have already conducted experiments on scenarios with low overlap, discussed the limitations of the Retrieval-then-Registration approaches (General Comment1), and proposed a novel end-to-end registration pipeline that is effective for the unbalanced point cloud pairs. We believe that this is an undiscovered area in the context of unbalanced point cloud registration in an end-to-end manner. We would appreciate the reviewer considered this when evaluating our work.
> > >
> > > We hope we have addressed the reviewer’s concerns based on our additional experiments and responses.
> > >
> > >
> > > [1] Link of figure: https://anonymous.4open.science/r/iclr2023-rebuttal-23CC/README.md

---

### Official Review · Reviewer_g5jK · 2022-10-25

**Confidence:** 5
**Clarity, Quality, Novelty And Reproducibility:** This work is clear, marginally novel …
**Correctness:** 2
**Technical Novelty And Significance:** 2
**Empirical Novelty And Significance:** Not applicable
**Recommendation:** 1

**Strength And Weaknesses:**

#Strenths#
This paper is well written and easy to follow.
#Weakness#
The major faults of this paper lie in the experiment part, which can be summarized as:
(1)	The starting point of this paper is inappropriate. Since this work considers the registration between a local 3D scan and a global 3D map, this experimental setup actually lies in the category of relocalization. However, no experiment is provided to compare the proposed UPPNet with existing relocalization methods.
(2)	The authors conduct extensive experiments to compare the proposed UPPNet with the pair-wise registration methods. However, the experimental setup in this paper is unfair and different from the original setting of pair-wise registration. Pair-wise registration is mainly used to align a captured 3D scan with another 3D scan instead of the reconstructed 3D map.
(3)	The authors mention the limitations of the paradigm of retrieval-to-registration in related work. However, there are no specific experiments to demonstration these claims. In fact, the proposed UPPNet is an end-to-end implementation of this paradigm and does not really addressed these limitations.

Additional comparative experiments are expected to demonstrate the motivation of this work, such as:
(1) Compare the registration performance of the proposed UPPNet with relocalization methods (such as DH3D [1]) rather than pair-wise registration methods.
(2) Conduct extensive experiments to compare the proposed UPPNet with the state-of-the-art combinations of retrieval-then-registration (such as MinkLoc3D + Geotransformer[2]).

In addition, in the implementation details in Appendix, the authors claim that the voxel size is set to 1m. I have two questions about this:
(1) Is this setting also the same on the indoor 3DMatch/3DLoMatch dataset? On the indoor dataset, only the registrations with RMSE < 0.2m are considered for the calculation of Registration Recall. If the voxel size is set to 1m, I think it is difficult to achieve such a high RR in Table 1.
(2) In original pair-wise registration papers, the values of voxel sizes on 3DMatch and KITTI are 0.03m and 0.3m, respectively. According to the visualization results, this setting is reasonable. But in this paper, in order to reduce memory consumption and improve computation efficiency, the voxel size is set to 1m, which not only loses a lot of geometric information, but also is extremely unfair to the pair-wise registration methods.

[1] Du J, Wang R, Cremers D. Dh3d: Deep hierarchical 3d descriptors for robust large-scale 6dof relocalization. ECCV, 2020.
[2] Qin Z, Yu H, Wang C, et al. Geometric transformer for fast and robust point cloud registration. CVPR, 2022.


**Summary Of The Paper:**

In this paper, the authors propose a neural architecture named UPPNet to solve the registration problem of a local 3D scan and the global 3D map. The proposed UPPNet mainly consists of three modules: submap matching, super-point matching, and point matching. The submap matching module aims to retrieval the most similar set of target points from the global 3D map for the source local scan. For the source local scan and the retrieved target point cloud, the super-point matching and point matching modules are progressively performed to achieve pair-wise registration.

**Summary Of The Review:**

This paper has major issues in the experimental part.

---

> ### Author Response · Authors · 2022-11-25
> **Our response**
>
> > Q1: Conduct extensive experiments to compare the proposed UPPNet with the state-of-the-art combinations of retrieval-then-registration (such as MinkLoc3D + Geotransformer[6]).
>
> A: We conducted additional experiments by combining the state-of-the-art point cloud retrieval method (MinkLoc3Dv2 [7]) and the state-of-the-art pairwise registration methods (Predator [2], CoFiNet [3], GeoTransformer [6]). Please refer to “General Comment Q2” for more detailed results and analysis. The results show that our method outperforms the Retrieval-then-Registration pipeline by a large margin.
>
> > Q2: Experimental setup in this paper is unfair and different from the original setting of pair-wise registration.
>
> A: We politely disagree with the comment. We evaluated our method under two different experimental setups for a fair comparison. (1) We use the **original setting** of pair-wise registration (Table 1 of main) for the previous balanced registration datasets (3DMatch, 3DLoMatch, and KITTI). Here, we **follow exactly the same evaluation setups** of [1, 2, 3, 4] (2) We use the **unbalanced setting** (Figure 3 of main) for the unbalanced datasets, i.e., KITTI-UPP and ScanNet. For the unbalanced setting, we **finetuned all baseline methods [1, 2, 3, 5] and ours on unbalanced datasets**. Please refer to Appendix A.3 for more details.
>
> As seen from Table 1 in our main paper, our model consistently outperforms the baselines [1, 2, 3, 4] under the original setting. Even in the unbalanced setting that aims to align a captured 3D scan and the reconstructed 3D map, our model still performs reliable registrations while other methods [1, 2, 3, 5] suffer from the point and spatial imbalance. These results, under two different experimental setups, reveal the robustness of the proposed method in **both cases** of the balanced setting (original) and unbalanced setting (ours, which is more challenging).
>
> > Q3: Proposed UPPNet is an end-to-end implementation of this paradigm and does not really addressed limitations of the paradigm of retrieval-to-registration in related work.
>
> A: As we stated in the “General Comment Q1”, we clarified the motivation of our work and the difference with the Retrieval-then-Registration pipeline. Contrary to previous retrieval [7] and registration methods [1, 2, 3, 4, 5, 6], the proposed model is able to learn strong features for the submaps and local geometries under spatial extent and point density imbalance, thus being able to solve the conventional **two-stage** problems with a **single-stage** approach. Please refer to “General Comment Q1” for more details.
>
>
>
>
>
> > Q4: Question about voxel size
>
> A: As we stated in Q2 and in Appendix A.3, we follow the previous literature [1, 2, 3, 4] for the training and evaluation of conventional pairwise registration datasets (3DMatch, 3DLoMatch, and KITTI). Specifically, we use a 2.5cm voxel size for the indoor datasets (3DMatch and 3DLoMatch) and 30cm for the outdoor dataset (KITTI). We set 1m voxel size for our KITTI-UPP dataset since the reconstructed map in KITTI-UPP has a large number of points, which is shown to be infeasible to all baselines (D3Feat [1], Predator [2], CoFiNet [3], HRegNet [5], and GeoTransformer [6]). Therefore, we use a 1m voxel size for both our approach and baseline approaches for a fair comparison. For training the pairwise registration method on unbalanced datasets, we do our best to search for the optimal hyperparameters for each method and evaluate the resulting methods with ours under exactly the same evaluation setup. We clarified this in our revised version of the paper. Please refer to Appendix A.3 for more details.

---

> > ### Author Response · Authors · 2022-12-12
> > **Response to Reviewer g5jk**
> >
> > Dear respectful reviewer, as suggested, we have conducted additional experiments and added our responses in “General Comment Q1 and Q2”. We kindly ask the reviewer to reconsider the evaluation of our submission, depending on how well our responses satisfy the reviewer’s major concerns about the motivation of our work and the difference between Retrieval-then-Registration pipelines.

---

> > > ### Comment · Reviewer_g5jK · 2022-12-13
> > > **reply to the authors after reading their responses**
> > >
> > > I'm glad to see the authors add those comparative experiments that we suggest. However, I think the motivation and experiments of this paper are still clearly flawed.
> > >
> > > #Motivation#
> > >
> > > 1)	Problem Definition
> > > It can be seen from the paper that the author intends to solve the 6DoF pose between a single LiDAR scan and a large-scale 3D map. In particular, the authors define this 6DoF pose estimation task as a new problem in pairwise point cloud registration. I disagree with the author’s standpoint, since this problem has been investigated, and actually lies in the category of relocalization, especially the same as the point cloud based 6DoF relocalization[1][2]. Additionally, according to the definitions in [1][2], the pairwise registration can be regarded as a subtask of the 6DoF relocalization. I think the authors would like to blur the boundary between the 6DoF relocalization and pairwise registration, but that's inappropriate.
> > >
> > > 2)	Methodology
> > > To realize the 6DoF relocalization, a typical approach in [1][2] is a two-step process: 1) coarse localization/retrieval using global descriptors, 2) precise 6DoF pose estimation by pairwise registration. According to the pipeline in Figure 1, it can be seen that the proposed UPPNet actually follows this paradigm. Specifically, the Submap Matching is equivalent to the first step, the Super Point Matching as well as Point Matching are equivalent to the second step, and the network structure of the latter is almost identical to the pairwise registration framework of CoFiNet. Of course, it is valuable that the authors combine the two steps and solve the problems with a single-stage approach in an end-to-end manner. Therefore, we recommend that the authors rewrite the motivation, because it is more appropriate to explain this issue from the perspective of the 6DoF relocalization.
> > >
> > > #Experiments#
> > >
> > > 1)	The KITTI-UPP Dataset
> > > The authors claim that the KITTI-UPP dataset is created by aggregating sequential LiDAR frames for each scene provided by the KITTI Odometry benchmark. Actually, the KITTI-UPP dataset is unnecessary, because many similar datasets have appeared in the point cloud based 6DoF relocalization field [2]. For instance, Komorowski et al. [2] aggregated multiple sequences in the KITTI Odometry dataset into a map to evaluate the performance of 6DoF pose estimation. Additionally, more alternative datasets can be found in loop closure [3][4] for LiDAR-based SLAM. In contrast, the KITTI-UPP dataset in this paper is not practical.
> > >
> > > 2)	Experimental Settings
> > > The authors should conduct more experiments on relocalization datasets, such as MulRan [5] and Apollo-SouthBay [6], rather than pairwise registration datasets. Besides, I still think the experimental setup in this paper is unfair to pair-wise registration, because the voxel size of 1m severely degrades the performance of those pairwise registration methods. An appropriate approach during the inference is to leverage the trained models released by the baselines and the original settings (the values of voxel sizes on 3DMatch and KITTI are 0.03m and 0.3m, respectively) to register the point cloud frame by frame, and then select the rigid transformation with the smallest registration error as the final result after traversing all frames. Such experimental setups are truly fair for the pairwise registration methods since all configurations are exactly the same as those original papers. I believe the final performance of pairwise registration methods will not be as bad as in this paper.
> > >
> > > 3)	Comparison with the combination of the state-of-the-art Retrieval and Registration methods
> > > For the pairwise point cloud registration methods, I guess the authors do not utilize the trained models released by the baselines and set the voxel size to 1m, which is unfair to the pairwise registration methods, as explained above. Additionally, if the combinations of retrieval-then-registration really have poor performance as illustrated by the authors, does it mean that 6DoF relocalization is really challenging on KITTI? Actually not. Because we found that EgoNN [2] achieves the success rate of 99.7% on the same evaluation metric, and other combinations of retrieval-then-registration also achieve satisfactory results above 93%. Therefore, we conjecture that there may be some problems in experiment parts of this paper, which cannot fully reflect the performance of the state-of-the-art Retrieval and Registration methods.
> > >
> > > [1] DH3D: Deep Hierarchical 3D Descriptors for 6DoF Relocalization
> > > [2] EgoNN: Egocentric Neural Network for Point Cloud Based 6DoF Relocalization at the City Scale
> > > [3] OverlapNet: Loop Closing for LiDAR-based SLAM
> > > [4] LCDNet: Deep Loop Closure Detection and Point Cloud Registration for LiDAR SLAM
> > > [5] MulRan: Multimodal Range Dataset for Urban Place Recognition
> > > [6] L3-Net: Towards Learning Based LiDAR Localization for Autonomous Driving

---

### Author Response · Authors · 2022-11-25
**General comment**

We thank all reviewers for their insightful comments and suggestions. For each reviewer, we provided responses to the corresponding review. We mainly clarify the motivation of our work and the major difference with the Retrieval-then-Registration pipeline (**“General Comment Q1”**). We also conducted additional experiments incorporating the retrieval method [7] and pairwise registration methods [2, 3, 6] (**“General Comment Q2”**). We have uploaded the revised version of the paper. We briefly summarize the updates as follows:

- We clarify the motivation of our work and the difference with the **Retrieval-then-Registration** pipeline.
- We compare our method with the Retrieval-then-Registration pipeline combining the retrieval method [7] and pairwise registration methods [2, 3, 6].
- We clarify the experimental setups on pairwise registration datasets, i.e., 3DMatch, 3DLoMatch, and KITTI, and unbalanced datasets, i.e., KITTI-UPP and ScanNet.
- We clarify the technical details of the proposed method, including the submap matching module and structured matching.


### **Q1 [g5jk, CTpR] : Motivation of our work & Difference with Retrieval-then-Registration pipeline**

The major difference between our work and previous registration [1, 2, 3, 4, 5, 6] or retrieval [7] methods is that **our method solves both subtasks in a single shot**.

As reviewers point out, conventional approaches handle registration and retrieval tasks sequentially. For example, given a query frame and a database of frames of a scene, (Stage #1) retrieval methods [7] aim to find the nearest frame from the database, i.e., **coarse-grained localization**, and then (Stage #2) registration methods [1, 2, 3, 4, 5, 6] align the query and the retrieved frame and predict the relative pose, i.e., **fine-grained registration.** However, the two-stage approaches have been tested on a dataset containing a huge number of ‘overlapping/adjacent’ frames. With this large database of frames, the retrieval stage must undergo **multiple forward passes** to embed multiple ‘overlapping/adjacent’ frames of the scene. For example, to retrieve the best matching frame from a scene of 4071 LiDAR frames in KITTI, retrieval networks perform 4071 forward passes to provide embeddings for all the input frames, i.e., 4071 global vectors of the input frames of **a single scene**.

In order to address the issue, we construct an entire map by aggregating overlapping frames and aim to directly align the reconstructed map and query frame (**unbalanced registration**). In doing so, we can embed the scene with a **single forward pass**, requiring only one forward pass to provide a single global descriptor of the entire scene. In our case, after embedding the scene with a single forward pass, submap proposal module partitions the embedded scene into multiple submaps, storing global vectors of submaps, e.g., 104 global vectors for a single scene of KITTI. In this experimental setup of unbalanced point clouds, e.g., registration of reconstructed map (the entire scene) and query, we empirically found that previous methods [1, 2, 3, 4, 5, 6, 7]  all underperform our method due to extreme imbalance of spatial extent and point density as shown in Figure 3 of our main paper.

In addition, we highlight that the conventional **two-stage** approaches have several limitations compared to our **single-stage** method.
1. The existing retrieval [7] and registration methods [1, 2, 3, 4, 5, 6] **suffer from point cloud imbalance issue** in terms of spatial extent and point density, thus struggle to generalize to the real-world scenarios, whereas ours surpasses them in the original experimental setup (as shown in Table 1 of main paper) as well as our unbalanced setup (as shown in Figure 3 of main paper). We also provide additional experiments in this rebuttal.
2. The two-stage methods **need to train two separate networks**, whereas ours is trained in an end-to-end manner, and as a consequence, our method is capable of learning more stronger features for the description of submaps and local geometries.
3. The **accuracy of pose estimation is bounded by the accuracy of the localization module** because the second stage depends on the predicted results in the first stage. In contrast, we adopt a multiple submaps strategy to guarantee the positive submap exists in the predicted top k submaps. Additionally, the proposed structured matching support the second stage not being significantly affected by localization results. We demonstrate the merit of our approach by providing an additional experiment shown in the next.

---

> ### Author Response · Authors · 2022-11-25
> **General comment (2)**
>
> ### **Q2 [g5jk, CTpR] : Comparison with the combination of the state-of-the-art Retrieval and Registration methods**
>
> As suggested by Reviewer g5jk and CTpR, we conducted additional experiments that are described as follows:
>
>
> **[Experiment #1]** First, we substitute our submap proposal module with MinkLoc3Dv2 [7], the state-of-the-art point cloud retrieval method, and evaluate the performance to clarify the efficacy of the proposed submap matching module. Please note that we choose MinkLoc3Dv2 [7] because it shows the best performance on 3D retrieval tasks outperforming DH3D [8] by a large margin. For a fair comparison, we train the MinkLoc3Dv2 [7] on our KITTI-UPP dataset for 200 epochs following the same train setups as ours.
>
> The results are reported in Tables 1 and 2. For the experiments with various point imbalance factors (Table 1), we fix the spatial imbalance factor, e.g., $\rho_s$ = 2.7. Likewise, we fix the point imbalance factor in Table 2, e.g., $\rho_d$ = 6.9. As shown in Tables 1 and 2, performance drops dramatically as the spatial and point imbalance factors become larger; due to the inaccurate submap matching results of MinkLoc3Dv2 [7].
>
> **[Table 1. MinkLoc3Dv2 + Ours under various point imbalance factors $\rho_p$]**
> | &nbsp; | $\rho_p$ | Recall (%) $\uparrow$  | TE (m) $\downarrow$  | RE (deg) $\downarrow$ | IR (%) $\uparrow$ |
> |:---|---:|---:|---:|---:|---:|
> | MinkLoc3Dv2 + Ours | 2.3 | 92.7 | 0.431 | 0.984 | 26.1 |
> | **Ours**  | &nbsp; | **99.3** | **0.359** | **0.762** | **20.9** |
> | MinkLoc3Dv2 + Ours | 3.7 | 84.0 | 0.490 | 1.213 | 25.3 |
> | **Ours**  | &nbsp; | **97.8** | **0.407** | **0.890** | **21.4** |
> | MinkLoc3Dv2 + Ours | 4.8 | 74.9 | 0.530 | 1.246 | 26.2 |
> | **Ours**  | &nbsp; | **95.6** | **0.412** | **0.902** | **21.7** |
> | MinkLoc3Dv2 + Ours | 6.9 | 65.5 | 0.631 | 1.356 | 21.4 |
> | **Ours**  | &nbsp; | **94.5** | **0.494** | **1.004** | **20.4** |
>
> **[Table 2. MinkLoc3Dv2 + Ours under various spatial imbalance factors $\rho_s$]**
> | &nbsp; | $\rho_s$ | Recall (%) $\uparrow$  | TE (m) $\downarrow$  | RE (deg) $\downarrow$ | IR (%) $\uparrow$ |
> |:---|---:|---:|---:|---:|---:|
> | MinkLoc3Dv2 + Ours | 2.3 | 92.7 | 0.431 | 0.984 | 26.1 |
> | **Ours**  | &nbsp; | **99.3** | **0.359** | **0.762** | **20.9** |
> | MinkLoc3Dv2 + Ours | 4.0 | 57.8 | 0.595 | 1.280 | 22.1 |
> | **Ours**  | &nbsp; | **94.2** | **0.445** | **1.042** | **20.6** |
> | MinkLoc3Dv2 + Ours | 6.3 | 38.2 | 0.639 | 1.411 | 25.5 |
> | **Ours**  | &nbsp; | **90.2** | **0.542** | **1.196** | **19.2** |
> | MinkLoc3Dv2 + Ours | 7.9 | 35.3 | 0.634 | 1.472 | 22.2 |
> | **Ours**  | &nbsp; | **86.2** | **0.579** | **1.327** | **18.5** |
> | MinkLoc3Dv2 + Ours | 11.1 | 30.9 | 0.708 | 1.607 | 14.5 |
> | **Ours**  | &nbsp; | **82.5** | **0.612** | **1.314** | **16.9** |

---

> > ### Author Response · Authors · 2022-11-25
> > **General comment (3)**
> >
> > **[Experiment #2]** Second, we combine the state-of-the-art retrieval method (MinkLoc3Dv2 [7]) with state-of-the-art pairwise registration methods (Predator [2], Cofinet [3], GeoTransformer [6]), and compare them with ours. We fix the spatial imbalance factor ($\rho_s$ = 2.7) and compare results in Table 3. Likewise, we fix the point imbalance factor ($\rho_d$ = 6.9) and show the results in Table 4. As reported in Tables 3 and 4, even though pairwise registration methods [2, 3, 6] benefit from MinkLoc3Dv2 [7], ours still outperforms them by large margins. We conjecture that the method of generating a global descriptor is not optimal for the challenging imbalance scenario. Note that MinkLoc3Dv2 [7] performs global average pooling that aggregates all the superpoint features by assigning equal weights to each feature to form a global descriptor. In contrast, our attention-based context aggregation module dynamically assigns weight values to each superpoint feature according to their importance, thus being able to provide more adaptive, reliable global descriptors for retrieval.
> >
> > **[Table 3. Combination of retrieval and registration under various point imbalance factors $\rho_p$]**
> >
> > |  | $\rho_p$ | Recall (%) $\uparrow$  | TE (m) $\downarrow$  | RE (deg) $\downarrow$ | IR (%) $\uparrow$
> > |:---|---:|---:|---:|---:|---:
> > MinkLoc3Dv2 + Predator  | 2.3 | 73.8 | 0.555 | 1.353 | 14.4
> > MinkLoc3Dv2 + CoFiNet  | &nbsp; | 82.5 | 0.630 | 1.436 | 11.1
> > MinkLoc3Dv2 + GeoTransformer  | &nbsp; | 93.5 | 0.309 | 0.591 | 11.6
> > **Ours**  | &nbsp; | **99.3** | **0.359** | **0.762** | **20.9**
> > MinkLoc3Dv2 + Predator  | 3.7 | 58.9 | 0.634 | 1.577 | 10.1
> > MinkLoc3Dv2 + CoFiNet  | &nbsp; | 67.3 | 0.731 | 1.744 | 10.2
> > MinkLoc3Dv2 + GeoTransformer  | &nbsp; | 89.5 | 0.321 | 0.560 | 11.9
> > **Ours**  | &nbsp; | **97.8** | **0.407** | **0.890** | **21.4**
> > MinkLoc3Dv2 + Predator  | 4.8 | 49.1 | 0.688 | 1.666 | 8.4
> > MinkLoc3Dv2 + CoFiNet  | &nbsp; | 58.2 | 0.773 | 1.821 | 10.0
> > MinkLoc3Dv2 + GeoTransformer  | &nbsp; | 82.2 | 0.336 | 0.653 | 11.5
> > **Ours**  | &nbsp; | **95.6** | **0.412** | **0.902** | **21.7**
> > MinkLoc3Dv2 + Predator  | 6.9 | 34.5 | 0.860 | 1.865 | 7.0
> > MinkLoc3Dv2 + CoFiNet  | &nbsp; | 44.7 | 0.864 | 1.897 | 10.2
> > MinkLoc3Dv2 + GeoTransformer  | &nbsp; | 77.5 | 0.377 | 0.733 | 10.1
> > **Ours**  | &nbsp; | **94.5** | **0.494** | **1.004** | **20.4**
> >
> >
> > **[Table 4. Combination of retrieval and registration under various spatial imbalance factors $\rho_s$]**
> >
> > |  | $\rho_s$ | Recall (%) $\uparrow$  | TE (m) $\downarrow$  | RE (deg) $\downarrow$ | IR (%) $\uparrow$
> > |:---|---:|---:|---:|---:|---:
> > MinkLoc3Dv2 + Predator  | 2.3 | 73.8 | 0.555 | 1.353 | 14.4
> > MinkLoc3Dv2 + CoFiNet  | &nbsp; | 82.5 | 0.630 | 1.436 | 11.1
> > MinkLoc3Dv2 + GeoTransformer  | &nbsp; | 93.5 | 0.309 | 0.591 | 11.6
> > **Ours**  | &nbsp; | **99.3** | **0.359** | **0.762** | **20.9**
> > MinkLoc3Dv2 + Predator  | 4.0 | 39.3 | 0.713 | 1.597 | 9.3
> > MinkLoc3Dv2 + CoFiNet  | &nbsp; | 40.7 | 0.763 | 1.816 | 9.2
> > MinkLoc3Dv2 + GeoTransformer  | &nbsp; | 66.5 | 0.355 | 0.742 | 9.8
> > **Ours**  | &nbsp; | **94.2** | **0.445** | **1.042** | **20.6**
> > MinkLoc3Dv2 + Predator  | 6.3 | 22.9 | 0.744 | 1.510 | 9.1
> > MinkLoc3Dv2 + CoFiNet  | &nbsp; | 28.4 | 0.798 | 1.796 | 9.9
> > MinkLoc3Dv2 + GeoTransformer  | &nbsp; | 45.1 | 0.488 | 0.787 | 8.7
> > **Ours**  | &nbsp; | **90.2** | **0.542** | **1.196** | **19.2**
> > MinkLoc3Dv2 + Predator  | 7.9 | 18.9 | 0.758 | 1.919 | 6.9
> > MinkLoc3Dv2 + CoFiNet  | &nbsp; | 21.5 | 0.741 | 1.990 | 9.3
> > MinkLoc3Dv2 + GeoTransformer  | &nbsp; | 45.8 | 0.450 | 0.811 | 8.6
> > **Ours**  | &nbsp; | **86.2** | **0.579** | **1.327** | **18.5**
> > MinkLoc3Dv2 + Predator  | 11.1 | 16.4 | 0.691 | 1.856 | 7.9
> > MinkLoc3Dv2 + CoFiNet  | &nbsp; | 20.7 | 0.842 | 1.957 | 9.2
> > MinkLoc3Dv2 + GeoTransformer  | &nbsp; | 40.0 | 0.475 | 0.883 | 7.9
> > **Ours**  | &nbsp; | **82.5** | **0.612** | **1.314** | **16.9**
> >
> > References
> >
> > [1] Bai et al., "D3feat: Joint learning of dense detection and description of 3d local features." CVPR 2020.
> >
> > [2] Huang et al., "Predator: Registration of 3d point clouds with low overlap." CVPR 2021
> >
> > [3] Yu et al., "Cofinet: Reliable coarse-to-fine correspondences for robust pointcloud registration." NeurIPS 2021
> >
> > [4] Choy et al., "Fully convolutional geometric features." ICCV 2019.
> >
> > [5] Lu et al., "Hregnet: A hierarchical network for large-scale outdoor lidar point cloud registration." ICCV 2021.
> >
> > [6] Qin et al., "Geometric transformer for fast and robust point cloud registration." CVPR 2022.
> >
> > [7] Komorowski, Jacek. "Improving Point Cloud Based Place Recognition with Ranking-based Loss and Large Batch Training." arXiv preprint arXiv:2203.00972 (2022).
> >
> > [8] Du et al., "Dh3d: Deep hierarchical 3d descriptors for robust large-scale 6dof relocalization." ECCV 2020.

---

### Decision · Program_Chairs · 2023-01-20

**Decision:**

Reject

**Justification For Why Not Higher Score:**

This paper receives 1x strong reject and 2x marginally above the acceptance threshold. However, the comments from all reviewers are mostly negative.


**Justification For Why Not Lower Score:**

NA

**Metareview: Summary, Strengths And Weaknesses:**

This paper receives 1x strong reject and 2x marginally above the acceptance threshold. However, the comments from all reviewers are mostly negative.

The 1x strong reject argues that: 1)  No experiment is provided to compare the proposed UPPNet with existing relocalization methods. 2) The experimental setup in this paper is unfair as pair-wise registration is mainly used to align a captured 3D scan with another 3D scan instead of the reconstructed 3D map. 3) The authors mention the limitations of the paradigm of retrieval-to-registration in related work, but the proposed UPPNet is an end-to-end implementation of this paradigm and does not really addressed these limitations.

Although there are 2x marginally above the acceptance threshold, these reviewers also feel that some important baseline experiments are missing, this brings a question -- can we combine the best practice of a state-of-the-art nearest frame retrieval method [e.g., https://github.com/jac99/MinkLoc3Dv2] and a state-of-the-art registration method [Geometric Transformer for Fast and Robust Point Cloud Registration] to solve the proposed task. This comment agrees with the concern from the strong reject reviewer. The other marginally above the acceptance threshold reviewer mentioned that there is a doubt about its method design. Moreover, many important details are missing or confusingly described in the paper.